# The Role of the Gut and Airway Microbiota in Chronic Rhinosinusitis with Nasal Polyps: A Systematic Review

**DOI:** 10.3390/ijms25158223

**Published:** 2024-07-27

**Authors:** Manuel Gómez-García, Emma Moreno-Jimenez, Natalia Morgado, Asunción García-Sánchez, María Gil-Melcón, Jacqueline Pérez-Pazos, Miguel Estravís, María Isidoro-García, Ignacio Dávila, Catalina Sanz

**Affiliations:** 1Institute for Biomedical Research of Salamanca (IBSAL), 37007 Salamanca, Spain; mgomezgarcia4.ibsal@saludcastillayleon.es (M.G.-G.); emorenoj.ibsal@saludcastillayleon.es (E.M.-J.); nmorgado@usal.es (N.M.); chonela@usal.es (A.G.-S.); mgilmel@saludcastillayleon.es (M.G.-M.); jperezpaz.ibsal@saludcastillayleon.es (J.P.-P.); misidoro@saludcastillayleon.es (M.I.-G.); idg@usal.es (I.D.); catsof@usal.es (C.S.); 2Pharmacogenetics and Precision Medicine Unit, Clinical Biochemistry Department, University Hospital of Salamanca, 37007 Salamanca, Spain; 3Department of Microbiology and Genetics, University of Salamanca, 37007 Salamanca, Spain; 4Biomedical and Diagnostics Sciences Department, University of Salamanca, 37007 Salamanca, Spain; 5Results-Oriented Cooperative Research Networks in Health—Red de Enfermedades Inflamatorias, Carlos III Health Institute, 28220 Madrid, Spain; 6Otorhinolaryngology and Head and Neck Surgery Department, University Hospital of Salamanca, 37007 Salamanca, Spain; 7Centre for Networked Biomedical Research in Cardiovascular Diseases (CIBERCV), Carlos III Health Institute, 28220 Madrid, Spain; 8Medicine Department, University of Salamanca, 37007 Salamanca, Spain; 9Department of Allergy, University Hospital of Salamanca, 37007 Salamanca, Spain

**Keywords:** chronic rhinosinusitis with nasal polyps, human microbiota, respiratory diseases, immune system, functional modulation of the microbiota

## Abstract

In recent years, there has been growing interest in understanding the potential role of microbiota dysbiosis or alterations in the composition and function of human microbiota in the development of chronic rhinosinusitis with nasal polyposis (CRSwNP). This systematic review evaluated the literature on CRSwNP and host microbiota for the last ten years, including mainly nasal bacteria, viruses, and fungi, following the PRISMA guidelines and using the major scientific publication databases. Seventy original papers, mainly from Asia and Europe, met the inclusion criteria, providing a comprehensive overview of the microbiota composition in CRSwNP patients and its implications for inflammatory processes in nasal polyps. This review also explores the potential impact of microbiota-modulating therapies for the CRSwNP treatment. Despite variability in study populations and methodologies, findings suggest that fluctuations in specific taxa abundance and reduced bacterial diversity can be accepted as critical factors influencing the onset or severity of CRSwNP. These microbiota alterations appear to be implicated in triggering cell-mediated immune responses, cytokine cascade changes, and defects in the epithelial barrier. Although further human studies are required, microbiota-modulating strategies could become integral to future combined CRSwNP treatments, complementing current therapies that mainly target inflammatory mediators and potentially improving patient outcomes.

## 1. Introduction

Chronic respiratory diseases (CRDs) are persistent and non-communicable diseases affecting the upper and lower airways that cause significant morbidity and mortality [1]. Chronic rhinosinusitis (CRS) is characterized by an inflammatory disorder involving the nose, paranasal sinuses, and upper airways, persisting for at least 12 weeks despite appropriate medical therapy. This debilitating disease affects 5–12% of the general population and common symptoms, which might be similar to acute cases, include anterior or posterior nasal discharge/congestion, facial pain or pressure, impairment of smell or anosmia, difficulty breathing through the nose, headache, fatigue, and cough [2,3]. Anatomical changes of the nose and paranasal sinuses contribute to the presence and recurrence of CRS symptoms. Imaging studies have shown a significant reduction in the size of the maxillary, sphenoid, and frontal sinuses, along with near-total opacification of these areas and the ethmoid air cells. There is also a rarefaction of the ethmoid bony septae, as well as the ostiomeatal units, and sphenoethmoidal recesses, and frontal outflow tracts are compromised due to thickened mucosa and retained secretions [4,5,6,7].

The CRS incidence has increased, leading to subsequent socio-economic impacts on healthcare systems and economies [8]. In the United States, handling CRS is projected to cost between USD 11 billion and USD 13 billion annually, equating to USD 2609 per patient per year. Meanwhile, CRS contributes to a direct healthcare spending increase of EUR 2500 per patient per year in Europe. Beyond direct expenses, indirect costs stemming from absenteeism and reduced workplace productivity are substantial, with CRS ranking among the top 10 most expensive health conditions for United States employers, surpassing USD 20 billion annually [9].

Although with significant limitations, CRS has traditionally been classified into two phenotypes based on diagnosis with nasal endoscopy and inflammatory biomarkers: CRS with nasal polyps (CRSwNP) and CRS without nasal polyps (CRSsNP) [2,10]. CRSwNP is typically characterized by nasal congestion and/or congestion and loss of smell and taste, while CRSsNP is mainly associated with rhinorrhea and facial pain or pressure [2,11]. Based on endotype dominance, CRSwNP is predominantly related to type 2 inflammatory response in the general adult population in Western countries. It is characterized by activation and recruitment of T-helper type-2 (Th2) cells, e.g., eosinophils and mast cells, as well as increased levels of Th2 inflammatory interleukins (ILs), e.g., IL-4, IL-5, and IL-13, and immunoglobulin E (IgE). In contrast, CRSsNP, which is more prevalent than CRSwNP [9,11], has been associated with non-eosinophilic/non-type-2 inflammation involving a mixture of type-1 and type-3 inflammatory response mediated by Th1 and Th17 cells, respectively [9,12,13,14]. Since type-2 inflammation is involved in the pathogenesis of several comorbidities, the prevalence of asthma, and/or nonsteroidal anti-inflammatory drug-exacerbated respiratory disease (NSAID-ERD) or aspirin-exacerbated respiratory disease (AERD) in patients with CRSwNP is higher than those with CRSsNP [15,16,17]. Consequently, CRSwNP patients often show more severe clinical symptoms, reduced quality of life (QOL), greater healthcare resource utilization, and direct and indirect costs, thus making this phenotype more clinically relevant [15,18,19,20].

The prevalence of CRSwNP in the general population has been estimated to range from 1 to 3%, although it varies across diverse regions of the world [16,17,21]. About the diagnosis, CRSwNP is confirmed by computed tomography scans and nasal endoscopic visualization of nasal polyps with specific scoring systems such as the Lund–Mackay score [22,23]. Nasal polyps or fleshy swellings result from soft tissue growth in the lining of the nose and paranasal sinuses, probably due to chronic inflammation of unknown causes [21]. They usually manifest as bilateral lesions, being more frequent in males (3.2–3.7%) than females (2.0–3.3%) and for the age group ranging from 60 to 69 years [19,21,24]. Besides male sex and aging, the other most commonly reported risk factors for developing CRSwNP include diseases associated with a worse respiratory function, such as asthma, eosinophilia, allergy, smoking, obesity, or variations and polymorphisms in genes related to immune response, structural proteins, or tissue remodeling [9,21,25,26,27,28]. The presence of microbiota dysbiosis or alterations in the composition, function, and local distribution of host microbiota also implies an increased risk for nasal polyps [29].

The human microbiota is defined as a community of microorganisms, including mainly a wide variety of bacterial species, as well as archaea (primitive single-celled organisms), eukarya, and viruses that live in and on the mucosal surfaces of the gut and skin, as well as the respiratory and genital tract [30]. Host–microbial community interactions have been determined to play a crucial role in developing and maintaining several vital functions, such as regulation of the immune system, protection against pathogens, digestion of food, biotransformation of foreign substances, or production of vitamins, among others [31,32,33]. However, older age; host genetic susceptibility; environmental factors; infections; and changes in antibiotic use, diet, or lifestyle can lead the transition from host–microbiota symbiosis to dysbiosis, contributing to the pathogenesis of a variety of inflammatory and autoimmune diseases, including CRS [29,34,35].

Accumulating data suggest that CRSwNP immunopathogenesis can be partially explained by alterations in the host microbiota, such as reductions in bacterial diversity with an increase in pathogenic bacteria that induce inflammatory reactions (e.g., *Staphylococcus aureus* or *Pseudomona aeruginosa*) and a loss of beneficial bacteria with potential immune protective properties (*Lactobacillus*, *Dolosigranulum*, and *Citrobacter* species). The formation of nasal biofilms, referring to microbial communities that adhere to a surface or biological tissue and are embedded in a matrix of self-produced extracellular polymeric substances, can also be relevant [36]. The presence of defects in the nasal mechanical or physical barrier with inappropriate immune response can also be a critical factor [29,37,38,39]. In this context, an enhancement of the permeability due to environmental and intrinsic factors of the nasal physical barrier, which consists of airway mucus, epithelial cells with cilia, adhesion molecules, or apical junctional complexes (tight and adherens junctions, desmosomes, and hemidesmosomes), as well as endogenous antimicrobial substances [37,40], was previously associated with CRSwNP [37,41,42,43]. A vulnerable epithelial barrier to pathogens and allergens/foreign material contributes toward activating an intense local Th2 response of the adaptive immune system with an accumulation of inflammatory cells, which triggers a remodeling of nasal mucosal tissue pattern consisting of decreased collagen and increased fibrin. This dynamic inflammatory process leading to polyp formation is described under the immune barrier hypothesis [29,37,39,42,44].

Considering the potential role of the host microbiota in CRSwNP and the estimation that up to 40% of CRSwNP patients cannot achieve an acceptable level of disease control despite adequate endoscopic sinus surgery (ESS), corticosteroid treatment, and up to two short courses of systemic corticosteroids in the last year (difficult-to-treat patients according to the European Position Paper on CRS (EPOS) guideline) [9,15], modulation of the nasal and gut microbiota emerges is a promising therapeutic strategy worth exploring. Besides ESS and oral and intranasal corticosteroids, the mainstays of treatment for patients with CRSwNP involve other medical management/medication, including antibiotics and antifungals, antihistamines, and monoclonal antibodies targeting interleukins, such as IL-4, IL-5, and IL-13, or IgE (benralizumab, dupilumab, mepolizumab, and omalizumab), among others with varying efficacy [9,21,43,45,46].

To take a deep look into the complex role of the human microbiota in the pathogenesis of CRSwNP, we conducted a systematic literature review of the last decade, focusing on the nasal and gut bacterial microbiota. The microbiome was also considered, which refers to the microorganisms and their genomes occupying a reasonably well-defined habitat and microbial structural elements, metabolites, and environmental conditions [47]. The results of this systematic review are organized into three main sections, which provide a concise and precise examination of the most recent publications focused on the composition of nasal and gut microbiota in CRSwNP patients and point out the critical microbial species associated with nasal polyps (“Microbiota Composition in CRSwNP”), on the potential mechanism underlying microbial dysbiosis and inflammatory processes in nasal polyps (“Relationship between Microbial Dysbiosis and Inflammation in CRSwNP”), and on the evaluation of promising therapeutic strategies involving the modulation of the bacterial microbiota for the CRSwNP treatment (“Effect of CRSwNP Treatment on the Modulation of the Microbiota”). The review includes both adult and children studies and excludes literature reviews.

## 2. Materials and Methods

### 2.1. Study Design and Search Strategy

It is worth noting that the Preferred Reporting Items for Systematic Reviews and Meta-Analyses (PRISMA) 2020 guidelines [48] and the Grading of Recommendations, Assessment, Development, and Evaluations (GRADE) recommendations [49] were taken into account to carry out this systematic review. PICO (Population, Intervention, Control, and Outcome) criteria [50] were defined as follows: P = CRSwNP, intervention = host microbiota or dysbiosis contributing to the CRSwNP pathogenesis; C = disease control or healthy subjects, placebo, or standard-care groups, or comparisons before or after the implementation of microbiota-related interventions; and O = outcomes related to the impact of microbiota on CRSwNP, such as changes in symptom severity, recurrence rates, quality of life, histological findings, or other clinically relevant measures.

The literature search was carried out between January and February 2024 in Medline (PubMed), Wiley (Cochrane Library), Embase, and Scopus online databases using a combination of terms related to “Chronic rhinosinusitis with nasal polyps” or “nasal polyps” and “microbiota” or “microbiome” or “dysbiosis”. The selected terminology was chosen considering MeSH and Emtree terms from Medline and Embase, respectively, incorporating input from experienced specialists in the field and reviewing relevant literature. Available government and professional organization websites were also manually reviewed, and a references list of included articles was used to avoid missing any relevant studies. Appendix A provides details of the search strategies.

### 2.2. Selection Criteria

All original articles and meta-analyses indexed from 2014 to February 2024 focused on the evaluation of the microbiota in CRSwNP were identified as eligible studies using the following inclusion criteria: (1) original study or meta-analysis written in English; (2) studies including human subjects, both children and adults; (3) patients with CRSwNP or nasal polyps; and (4) studies describing key microbial species, dysbiosis, or microbiota changes in associations with disease onset, severity, prevalence, or treatment. In contrast, the exclusion criteria were (1) animal, in vitro or in silico studies; (2) review articles, case reports, expert-opinion articles, commentaries, or letters; (3) conference abstract published after 1 January 2023; (4) articles mainly focused on other diseases, such as antrochoanal polyps; and (5) articles whose full-text version was not available to us.

### 2.3. Review Process and Data Extraction

Three review authors carried out the initial screening of abstracts and titles retrieved from each study, using the open-access Rayyan software [51]. The consideration of each study for full review was based on the PICO previously described, when it was possible. Subsequently, at least two reviewers independently screened the full-text articles selected. Any disagreements during screenings were resolved by consensus or a third independent reviewer.

Data reported in the selected articles were collected using a standardized data-extraction template in Microsoft Excel, allowing us to plot the information for each of this review’s previously described main sections. The following information was extracted: information about the article (name of the first author, years of publication, and title) and the study (study type, objective, population/country in which the study was carried out, and sample size), patient demographics (age, sex, and other diseases associated/comorbidities), details of microbiota-related interventions (type/format), quantitative, and/or qualitative results and conclusions.

A risk of bias (RoB) assessment was carried out to establish the degree of certainty and quality of the randomized controlled studies, which included using version 2 of the Cochrane risk-of-bias tool (RoB 2) [52]. In detail, studies were classified as “High, Moderate, and Low RoB”. As in previous studies, the Newcastle–Ottawa Scale (NOS) tool for cohort and case–control studies [52,53] and an adapted form of NOS for cross-sectional [54] were used to assess the validity of the findings of non-randomized studies. The scales assess aspects related to study quality concerning selection (four points for cohort and case–control studies or five points for cross-sectional studies) and comparability (two points) of study groups, as well as assessment of outcomes or ascertainment of exposure (three points) [53]. NOS scores were again classified as “High” (zero-to-three NOS points for cohort and case–control studies and zero-to-four points for cross-sectional studies), “Moderate” (four-to-six or five-to-seven points), and “Low” (seven-to-nine or eight-to-ten points) RoB. It is worth noting that a lower score was assigned to any study that referred to too-small sample size (less than ten CRSwNP patients), was without appropriate controls, or did not clearly describe the methodology used.

## 3. Results

### 3.1. Search Results

Before the beginning of this systemic review, the review protocol was registered on the International Prospective Register of Systematic Reviews (PROSPERO, study ID: CRD42024509276).

The initial database search using synonymous terms of CRSwNP, nasal polyps, microbiota, microbiome, and dysbiosis yielded 464 references that initially met the inclusion criteria previously described. After identifying and excluding duplicates, 300 references were selected for title and abstract review, and, subsequently, 111 references were selected for a full-text review. As a result, a total of 70 eligible studies (65 articles published and 5 included after manual searching) that met the inclusion criteria were finally included in this review. Figure 1 summarizes the results of each screening stage and the selection of studies presented as a PRISMA flow diagram.

Out of 70 studies included in this systematic review (Appendix A), 68 (97.4%) focused on the identification of the microbiota composition of CRSwNP subjects, 21 (30.0%) on the study of the relationship between microbial dysbiosis and inflammation in CRSwNP, and 9 (12.9%) on the evaluation of the effect of different CRSwNP treatment on the microbiota. Note that the studies have a publication range from 2014 to 2024, with 60 studies published relatively recently, between 2019 and 2024 (60.0%). The most significant number of studies was from Asia (24, 34.3%, mainly from China), and to a lesser extent from Europe (14, 20.0%), Oceania (13, 18.6%, mainly from New Zealand), and North America (13, 18.6%, mainly from the USA). Only one study included was a multicenter international cohort study across nine countries [55].

The studies collected data from more than 7450 subjects, of whom approximately 4000 were CRSwNP patients ranging in age from 13 to 88 years. Forty-nine studies (70%) were relatively small (initial sample size less than 100 subjects), and only ten studies showed results from more than 100 CRSwNP patients (14.3%). Considering the studies that reported the gender data, men (2665 subjects of whom 2103 were patients with CRSwNP) were more abundant than women (1451 subjects, 927 patients with CRSwNP). Disease-control (non-CRS) and/or healthy subjects were also included in 41 studies (58.6%), and CRSsNP patients in 39 studies (55.7%). Seven studies distinguished CRSwNP patients as eosinophilic (ECRSwNP) and non-eosinophilic (NECRSwNP) based on the infiltration of eosinophils in inflamed tissues (proportion of eosinophils more than ten percent and less than ten percent of inflammatory cells, respectively) [56,57,58,59,60,61,62,63]. Comorbidities, such as atopy, allergy, allergic rhinitis, aspirin sensitivity, acetylsalicylic acid sensitivity, AERD, and asthma—the most commonly reported—were also included in 49 studies (70%). A total of 26 studies (37.1%) reported that all subjects or a majority of them had not taken antibiotics and/or corticosteroids—two of the most commonly used drugs for CRSwNP [21]—at least the month before the analysis.

Regarding the study design, most of them (64, 91.4%) were identified as non-randomized studies of frequency (cohort or cross-sectional designs) or exposure (cohort and case–control studies), with only 6 (8.6%) being randomized controlled trials. In detail, 52 (74.3%) cross-sectional, 10 (14.3%) cohort, and 2 (2.9%) case–control studies were included in this review. Considering the prospective studies (one randomized controlled trial and nine cohort studies) that reported the follow-up time, the average follow-up was 10.4 months, with the shortest and longest follow-up time of two weeks and five years, respectively.

As previously described, the current version of the RoB 2 was used to performed a RoB assessment for randomized controlled trials and NOS tools to assess the validity of the findings of non-randomized controlled studies. The main concern regarding the RoB assessment referred to a well-defined selection of CRSwNP patients, a justified and satisfactory sample size, appropriate controls, or healthy subjects, or the description of the methodology selected to achieve the intended results. Considering that the RoB 2 tool showed that one (16.7%) randomized controlled trial was classified as “Low RoB” and four (66.7%) and “Some concerns”, mainly referred to the randomization process and deviations from intended interventions. One study (16.7%) accomplished the criteria for “Low RoB” because there were deviations from the intended intervention which arose because of the experimental context and some concerns about the randomization process (Figure 2a and Appendix A). By using NOS questionnaires, 44 of 64 (68.8%) non-randomized controlled studies included were considered “High quality”, whereas 17 (26.6%) and three (4.7%) met the criteria of “Moderate and Low quality”, respectively (Figure 2b and Appendix A). Of note, 27 (42.2%) studies did not include disease (non-CRS) or healthy control subjects, 2 (3.1%) had no reference to the number of CRSwNP patients included, and 7 (10.9%) referred to less than ten CRSwNP patients. All of them described the methodology used. Overall, the assessment of outcomes or ascertainment of exposure was the better-scored category.

### 3.2. Study Analysis

#### 3.2.1. Microbiota Composition in CRSwNP

Table 1 summarizes the most relevant information extracted from the studies included in this systematic review and focuses on the microbiota composition analysis in CRSwNP patients. Out of 66 studies, 61 (92.4%), 9 (13.6%), and 3 (4.5%) studies reported data about the bacterial, fungal, and viral composition from nasal samples, respectively, mainly taken from middle meatus (41, 62.1%). Only one study also focused on the gut microbiota from fecal samples [61]. Of note, two articles showed the same results of nasal microbiota composition obtained for the same CRSwNP patients [64,65].

The microbiota composition analyses were carried out using both molecular techniques (47, 71.2%), based mainly on marker-gene sequencing analysis with clustering of the sequencing reads into operational taxonomic units (OTUs) at a fixed similarity threshold (>97%), as well as standard microbiological techniques (40.9%), such as bacterial culture, typical colony morphology analysis, Gram staining, or biochemical test, among others.

Regarding the results for bacteria, it is worth noting that six out of seven studies with culture-positive patient data showed that the positive-culture rate was relatively high in CRSwNP patients (over 63.0%) [38,57,58,66,67,68,69]. The three studies focused on the characterizing of bacterial biofilm on the nasal mucosa in CRSwNP patients also showed a prevalence of over 44.2% [70,71,72]. Overall, the bacterial composition of the core microbiota found was dominated by the phylum *Firmicutes*, highlighting the bacterial genera *Staphylococcus* and *Streptococcus*, followed by *Proteobacteria*, with the genera *Haemophylus* and *Enterobacter*; and *Actinobacteria*, with *Corynebacterium* and *Propionibacterium* (*Cultibacterium*). The most frequently identified bacterial species were *Staphylococcus aureus*, coagulase-negative staphylococci (CoNS) or *Staphylococcus epidermidis*, *Streptococcus pneumoniae*, *Escherichia coli*, *Pseudomona aeruginosa*, *Enterobacter aerogenes*, *Klebsiella neumoniae*, and *Propionibacterium acne* (*Cultibacterium acne*) (Figure 3).

In the case of the results for fungal microbiota, different studies showed low isolation or identification of fungi [38,72,73,74,75]. All fungi reported were from the phyla *Basidiomycota* or *Ascomycota*, highlighting the following fungal genera: *Aspergillus*, *Malassezia*, and *Candida* [73,74,76,77,78,79]. About the few results obtained for the virus [80,81,82], note that human herpesvirus 6 (HHV-6), Epstein–Barr virus (EBV), cytomegalovirus (CMV), and Herpes simplex virus type 1 (HSV-1) showed the highest prevalence in 21 CRSwNP patients [80]. Significantly elevated IgG and IgA antibodies to influenza A virus (H3N2 and H1N1) and rhinovirus were also identified in CRSwNP patients [82].

Out of 24 studies that compared the results obtained in CRSwNP patients with those in disease control patients (non-CRS) or healthy subjects, 6 studies reported the lack of significant differences in the bacterial species or the number of different isolates obtained [73,83]; the total bacterial load [24]; both the alpha diversity (intra-community diversity mainly assessing through Chao1 index, focused on total species richness, and the Shannon index, assessing evenness in addition to richness) and beta diversity (intersample variability) [84]; the mycobiome composition [75]; or the whole microbiome identified [85]. The results shown in the rest of these studies are controversial regarding data interpretation. However, the most common differences reported were related to a decrease in alpha diversity [24,60,61,75,86,87,88,89], along with an underabundance of the bacterial genera *Corynebacterium* [55,64,65,86,88,90] and *Dolosigranulum* [64,65,84,86], as well as an overabundance of different bacteria of the phylum *Proteobacteria*, such as *Haemophylus*, *Moraxella*, or *Pseudomonas* [24,38,40,88,89,91]. Differences between both groups in the abundance or prevalence of specific *Firmicutes* species were also found [24,40,55,60,89,90]. The only study that compared the fungal microbiota of sinus mucosa between CRSwNP patients and control subjects obtained a significantly greater abundance of *Alternaria* and *Ranularia* and a significantly lower abundance of *Cutaneotrichosporon* in CRSwNP patients than in controls [75].

The studies evaluating the sinus microbiota of ECRSwNP and NECRSwNP patients showed differences in bacterial composition mainly at the genus level [59,60,61,62,63,86]. Remarkably, a higher abundance of *Moraxella* was found in the ECRSwNP group [60,61,63]. Likewise, prospective cohort studies through the post-operative follow-up of CRSwNP patients for one or five years showed no significant differences in microbial diversity between patients with recurrence or non-recurrence of nasal polyps (NPs). However, a notable variation in the abundance of specific bacterial genera was also observed [64,92].

Finally, significant differences in the nasal microbiota composition at different sample types but not at sample sites have been reported [90,93,94]. Gender-specific differences were also found in CRSwNP patients who used intranasal corticosteroid spray, obtaining a significantly higher abundance of *Corynebacterium* in men than in females [95]. Three studies showed that the bacterial spectrum changed after functional endoscopic sinus surgery (FESS) [38,79,96], and specific bacterial genera were significantly correlated with the disease severity measured using the Lund–Mackay score in computed tomography (CT) scans [65,86,94]. There was controversy about the relationship between the microbiota and clinical characteristics of CRSwNP patients, such as asthma, allergy, and sensitivity to drugs [24,38,97,98].

**Table 1 ijms-25-08223-t001:** Summary of the main findings of the included studies focused on the microbiota composition analysis in CRSwNP patients.

Ref.	Study Type	Objective	Sample Size	Country	Age (Average)	Sex (n with CRSwNP)	Other Disease (n with CRSwNP)	Sample Type	Methodology	Results/Conclusion	Observ.
[73]	CSS	To compare the predominant bacteriological profiles in the middle meatus of Chinese CRSwNP and CRSsNP patients and DC subjects.	537 (285 completed the study) CRSwNP: 165 CRSsNP: 76 DC: 44	China	18–80 CRSwNP: 18–78 (45.2)	M: 185 (109 with CRSwNP) F: 100 (56)	AS: 14 (13 with CRSwNP) AL: 11 (10) AsS: 1 (1)	MMSS collected during ESS	SMT	The most common bacteria were Coagulase-negative staphylococci (CoNS), *Streptococcus*, *Corynebacterium* spp., *Staphylococcus aureus*, and *Haemophilus influenzae*. No significant differences in the species or the number of different bacterial isolates between CRSwNP and DC. Very few fungi were isolated: one case each of *Penicillium notatum*, *Verticillium*, *Aspergillus*, and *Schizophyllum*.	No antibiotics or corticosteroids in the previous two weeks
[80]	CS	To evaluate human herpesviruses (HHV) 1–6 and community-acquired respiratory viruses (CARVS) prevalence in CRSwNP patients undergoing FESS	35 CRSwNP: 35	Italy	CRSwNP: 23–77 (50.3)	M: (25) F: (10)	AS: (8) AL: (15)	Bioptic samples of NP collected during FESS	MT (PCR and qPCR)	60% of patients (21/35) were positive for at least one virus in at least one specimen studied. The highest prevalence was found for herpesviruses 6 (HHV-6), Epstein–Barr virus (EBV), cytomegalovirus (CMV), and Herpes simplex virus type 1 (HSV-1).	NA
[96]	RCT	To evaluate the efficacy of long-term antibiotic therapy to prevent recurrence of NP	66 CRSwNP: 66 (55 completed the study)	Russia	18–77 (48.7)	M: 36 F: 30	AS: 35 AT: 19 AERD: 27	MMSS pre-FESS and post-FESS	SMT (system Microscan walk away 40)	The most common bacteria were *S. aureus*, *Streptococcus haemolyticus*, *Escherichia coli*, *Pseudomonas aeruginosa*, and *Enterobacter aerogenes*. The bacterial spectrum changed significantly after surgery.	No corticosteroids
[66]	CS	To evaluate the association between bacterial infection and surgical outcomes following FEES	71 CRSwNP: 71	Republic of Korea	NA	M: 42 F: 29	* Patients with a history of asthma were excluded	Maxillary sinus samples after MM antrostomy	SMT	Most of the patients (55/71, 77.5%) showed positive culture results. The most common bacteria were *S. epidermidis*, *Propionibacterium acne*, *Corynebacterium*, and *E. aerogenes*. The “normal flora group” (*S. epidermidis* and *Corynebacterium*) had the worst prognosis of postoperation, while the “culture-negative group” had the best prognosis.	NA
[99]	CSS	To compare the diversity of nasal microbiota and their secreted extracellular vesicles between CRSwNP patients and DCs	11 CRSwNP: 5 CRSsNP: 3 DC: 3	Republic of Korea	19–67 (43.1) CRSwNP: 23–67, (45.6)	M: 8 (4) F: 3 (1)	AT: 3 (1)	Nasal lavage fluid samples during ESS	MT (sequencing, OTUs)	The major bacterial genera were *Pseudomonas*, *Haemophilus*, *Staphylococcus*, *Moraxella*, *Enterobacter*, and *Fusobacterium* *.	No antibiotics or corticosteroids in the previous month
[100]	CSS	To determine whether smoking affects CT score, bacterial diversity of the upper airways, and distribution of inflammatory cells in nasal mucosa in CRS patients	64 CRSwNP: 20 CRSsNP: 17 DC: 27	Slovakia	CRSwNP: 22–76 (49.6)	NA	Patients with a history of asthma, atopy, and aspirin intolerance were excluded	MMSS	SMT	The most common bacteria were CoNS, *S. aureus*, and *Corynebacterium* species.	No antibiotics in the previous month
[70]	RCT	To compare the efficacy of topical nasal corticosteroids either as monotherapy or combined in eradicating the biofilm of NP	44 CRSwNP: 44 (34 completed the study)	Turkey	NA	M: 26 F: 18	NA	NP	Scanning electron microscopic (SEM) examination	In the initial baseline SEM examination, biofilm prevalence was 68.0% (23/34).	No antibiotics and/or corticosteroids in the previous month
[81]	CSS	To determine the presence of respiratory viruses in the paranasal sinuses of CRSwNP and CRSsNP patients compared to DCs	35 CRSwNP: 13 CRSsNP: 8 DC: 14	USA	CRSwNP: (55.3)	M: 19 (9) F: 16 (4)	AS: 15 (11) AR: 19 (11) AERD: 2 (2)	MMSS	MT (RT-PCR)	The presence of viruses (Coronavirus) was detected in only 7.7% of patients (1/13). None of the controls had a positive screen.	NA
[71]	CCS	To determine the presence of bacterial biofilm on the sinus mucosa of CRSwNP and CRSsNP patients and DCs	100 CRwNP: 18 ACP: 12 CRSsNP: 20 DC: 50	India	NA	NA	NA	Nasal–sinus tissue sample	SMT	Bacterial biofilm was detected in 66.7% of patients (12/18).	NA
[97]	CSS	To determine whether Gram-negative bacterial carriage impacted disease evolution and inflammatory profile in CRSwNP patients	337 CRSwNP: 337	Canada	NA (approx. 50)	NA	AS: aprox. 200 AL: 215 ASA: 101 (self-reported)	MMSS	SMT	The most common bacteria were CoNS, *S. aureus*, and *C. diphtheriae*. *P. aeruginosa* carriage was associated with a higher self-reported incidence of asthma.	NA
[95]	CSS	To evaluate the impact of saline irrigation and topical corticosteroids on the post-surgical sinonasal microbiota of CRSwNP patients and DCs	42 CRSwNP: 14 DC: 28	USA	22–77 CRSwNP: 27–65 (52.0)	M: 15 (3) F: 27 (11)	NA	Nasal cavity and maxillary swab samples	MT (sequencing, 16S RNA gene sequences)	The most abundant bacteria were *Propionibacterium*, *Corynebacterium*, *S. aureus*, *S. epidermis*, and *Staphylococcus pasteuri.* A higher proportional abundance of *Corynebacterium*, *Serratia*, and *Finegoldia* was found in men than in women who used intranasal steroid spray.	No antibiotics or corticosteroids in the previous two months
[101]	CSS	To determine the association between smoking history and sinonasal microbiome alterations in both CRS patients and DCs	101 CRSwNP: 22 CRSsNP: 48 DC: 31	USA	>18	M: 59 F: 42	AS: 31 AR 60	MMSS collected during ESS	MT (PCR and sequencing, OTUs)	The most abundant genera were *Staphylococcus*, *Corynebacterium*, *Carnobacteriaceae*, and *Streptococcus*. Smoking had a stronger effect on the microbiota at the phylum level (*Bacteroides*, *Firmicutes*, and *Proteobacteria*). Cigarette-smoking history was associated with less bacterial diversity, with a significant decrease in the phylum *Firmicutes* and an increase in the genera *Carnobacteriaceae*.	No systemic corticosteroids for at least one month
[76]	CSS	To determine the presence of specific fungal microbial species in CRS patients and DCss	28 CRSwNP: 15 Allergic fungal rhinosinusitis: 3 Fungus ball: 3 DC: 7	USA	CRSwNP: (49.0)	M: 17 (9) F: 11 (6)	AS: 12 (10) AsS: 4 (4)	Ethmoid and maxillary sinus brush samples	MT (qPCR)	*Malassezia* spp., *M. restricta*, and *M. globosa* were identified in ten, seven, and four CRSwNP patients, respectively.	NA
[67]	CSS	To define the bacteriology of CRSwNP and CRSsNP patients and DCs	163 CRSwNP: 60 CRSsNP: 50 DC: 26 HC: 27	Israel	CRSwNP: (51.2)	M: (37) F: (23)	NA	MMSS pre-FESS	SMT	Positive cultures (52/60) mainly for pathogenic bacteria (47/60) were found in most of the patients (86.7% and 78.3%, respectively). A higher rate of Gram-negative bacteria isolates than Gram-positive bacteria was found. The pathogenic bacteria most frequently isolated were *S. aureus*, *Citrobacter diversus*, *Proteus mirabilis*, *Enterobacter*, *Pseudomonas*, *Klebsiella oxytoca*, *S. pneumoniae*, and *Klebsiella pneumoniae.*	No antibiotics in the previous month
[68]	CSS	To define the bacteriology of CRSwNP and “sinonasal complication of dental disease or treatment” (SCDDT) patients	44 CRSwNP: 16 SCDDT: 28	Italy	CRSwNP: (49,4)	M: 26 (13) F: 18 (3)	NA	Endosinusal pus and biopsies from nasal polyps and fungus balls collected during surgery	SMT and MT (sequencing, 16S RNA gene sequences)	56.3% of patients (9/16) did not show microbial growth. *S. aureus* and other staphylococci, *Peptostreptococcus* spp., *E. coli*, and *Bacteroides* spp., were identified.	NA
[93]	CSS	To compare the nasal microbiota in regard to health state, anatomical region, and sampling strategy	79 CRSwNP: 15 CRSsNP: 27 DC: 37	Germany	18–79 CRSwNP: 20–77 (52.0)	M: 50 (13) F: 29 (2)	AS: 6 (4) AR: 23 (5) AsS: 3 (3)	Swab samples at four different regions along the nasal passage (anterior and posterior vestibules and inferior meatus and MM) and tissue biopsies	MT (sequencing, 16S RNA gene sequences)	No significant differences in global bacterial profiles at the sampling sites. Significant differences between the bacterial assemblages and diversity measured for different sample types (tissue biopsies and swabs).	No antibiotics at the time of sampling
[102]	CSS	To determine the association between the sino-nasal microbiota and toll-like receptor (TLR) activation in DCs and CRS patients	36 CRSwNP: 11 CRSsNP: 9 CRS/CF: 6 DC: 10	New Zealand	18–84 (43.9) CRSwNP: 18–71 (41.1)	M: 14 (7) F: 22 (4)	AS: 2 (8) AsS: 2 (0)	Nasal lavage and sino-nasal mucus samples collected during ESS	MT (qPCR and sequencing, OTUs)	*Pseudomonas*, *Haemophilus*, *Enterobacter*, and *Staphylococcus* were the dominating genera.	No antibiotics and prednisone
[103]	CSS	To determine the association of distinct pathogenic sinus microbiota with specific innate and adaptive immune responses and the relative risk of NP	76 (69 completed the study) CRSwNP: 32 CRSsNP: 27 DC: 10	USA	18–88 (46.6) CRSwNP: 19–88 (48.4)	M: 40 (21) F: 29 (11)	AS: 18 (12) CF: 9 (7)	Sinus brushing samples collected during ESS	MT (qPCR and sequencing, OTUs)	Most of the patients clustered into a subgroup typically dominated by *Corynebacteriaceae*	Most of patients (30/32, 93.8%) had taken pre-operative antibiotics (<3 months)
[90]	CSS	To compare the microbiota of the MM and inferior meatus in HCs, AR, and CRS patients and characterize intra- and inter-subject and inter-group differences	65 (48 for mapping intrasubject microbiota diversity and composition) CRSwNP: 18 CRSsNP: 15 AR: 11 HC: 4	USA	NA *	NA *	NA	MM and inferior meatus swab samples	MT (sequencing, OTUs)	No differences in phylogenetic diversity or Shannon diversity between MM and inferior meatus-associated microbiota. No differences in beta diversity across all subjects. Taxa enriched included *Staphylococcus* and *Alloiococcus*, as well as low-abundance *Corynebacterium*, *Haemophilus*, *Prevotella*, and *Porphyromonas* compared to HC.	No antibiotics and/or corticosteroids in the previous month
[74]	CSS	To analyze and quantify the sinonasal mycobiome in HC and CRS patients in an attempt to better elucidate its role in sinus disease	90 CRSwNP: 31 CRSsNP: 32 DC: 27	Australia	NA	NA	AS and AL	MMSS intra-operatively collected	SMT and MT (sequencing, OTUs)	Fungi were detected in 12.9% of patients (4/31) (*Aspergillus*, *Fusarium*, and without fungus identified).	NA
[104]	CSS	To study the association between inflammatory cells and signaling markers of CRS endotypes and the sinonasal bacterial community patterns	110 CRSwNP: 46 CRSsNP: 46 CRS/CF: 7 DC: 17	New Zealand (Most of the patients were European (31/38))	18–84 CRSwNP: 18–71 (48.0)	M: 55 (25) F: 55 (14)	AS: 48 (28) AsS: 14 (40) AERD: 11 (11)	Tissue biopsies collected from the bulla ethmoidal	MT (qPCR and sequencing, OTUs)	Patients were grouped in subject clusters mainly associated with *Staphylococcus*, *Corynebacterium*, *Streptococcus*, and *Propionibacterium*.	Most of the patients (35/39, 89.7%) had not taken pre-operative antibiotics and corticosteroids (<3 months)
[105]	RCT	To describe the effect of oral antibiotics and corticosteroids on the bacterial microbiome within the paranasal mucus in CRS patients	26 CRSwNP: 13 CRSsNP: 13	New Zealand	22–67 (48.9) CRSwNP: 29–64 (49.5)	M: 13 (7) F: 13 (6)	NA	MMSS	MT (qPCR and sequencing, OTUs)	Bacterial communities were typically dominated by *Corynebacterium* and *Staphylococcus* and at lower abundance by *Streptococcus*, *Dolosigranulum*, *Haemophilus*, and *Moraxella*.	No antibiotics and systemic corticosteroids in the previous month
[106]	CCS	To detect bacteria in culture-negative cases of CRS using 16S rRNA gene PCR and sequencing	20 CRSwNP: 15 CRSsNP: 5	India	NA	NA	NA	Discharge from the sinus and mucosal biopsies from the MM region collected during FESS	SMT and MT (qPCR and sequencing)	Two patients with a history of a previous sinus surgery yielded *Staphylococcus* spp. by qPCR (2/15, 13.3%). MRSA was isolated from one of them.	Cases with a history of a previous sinus surgery also had history of prior treatment with corticosteroids
[38]	CCS	To compare the microbiological features in middle meatus samples from CRSwNP and CRSsNP patients and DCs	251 CRSwNP: 136 CRSsNP: 66 DC: 49	China	CRSwNP: (45.4)	M: 154 (89) F: 96 (47)	AS: 41 (37) AR: 85 (61)	MMSS pre-ESS	SMT	Most patients (120/136, 88.4%) showed a positive culture result. The most abundant bacteria were CoNS, *Corynebacterium*, and S. *epidermidis*. The isolate rate of fungi was very low (3.7%). The strains were mainly Gram-positive aerobic and facultative anaerobic bacteria (69.8%). Patients with asthma showed a significantly lower isolation rate of *Corynebacterium* and *P. aeruginosa*. Patients with a history of ESS exhibited a significantly lower isolation rate of CoNS, and a significantly higher isolation rate of *P. aeruginosa*. A relatively high proportion of *Citrobacter* was observed compared with DCs. The isolation rate of *S. aureus* in the subgroup of patients with an increased percentage of eosinophils (>5%) in peripheral blood was higher than that in the subgroup with a standard percentage of eosinophils.	No antibiotics and glucocorticoids at least one month before surgery
[72]	CS	To investigate the relevance of the bitter-taste receptor TAS2R38 genetic variants in the susceptibility to bacterial infections associated with in vivo biofilm formation in CRSwNP patients	100 CRSwNP: 100	Italy	CRSwNP: (53.0)	M: (68) F: (32)	NA	Sinonasal mucosa samples pre-FESS	SMT and confocal laser scanning microscopy assay	63.0% of patients (63/100) showed positive culture result. The most common bacteria were *S. epidermidis*, *S. aureus*, and *Enterobacteriaceae* (*Klebsiella* spp., *Citrobacter koseri*, and *Serratia marcescens*). Only in one sample was found a mixed microbiota composed of *Candida albicans* with *S. aureus*. 19 of 43 samples (44.2%) were biofilm-positive. Biofilms were associated with *Klebsiella*, *Citrobacter*, *Haemophilus*, *Kocuria*, *S. aureus*, and *S. epidermidis*.	No antibiotics or corticosteroids in the previous month
[85]	CSS	To evaluate the bacterial community composition on the distinct types of CRS compared to healthy bacterial communities	18 CRSwNP: 5 CRSsNP: 5 CRS with unilateral purulent maxillary: 5 DC: 3	Germany	13–>70 CRSwNP: 13–29	M: 8 F: 10	NA	MMSS and tissue samples collected during ESS	SMT and MT (sequencing, OTUs)	Enterobacteria, Staphylococci, coryneform bacteria, Propionibacteria, Viridans streptococci, and *Haemophilus* were identified. The most common bacteria were *S. epidermidis*, *P. acnes*, and *Corynebacterium* spp. No significant differences were found in the microbiome between patients and DCs.	All patients had eosinophilia (>5%). No antibiotics in the previous month
[24]	CSS	To compare the bacterial communities of HCs with CRSwNP patients with (CRSwNP+A)/without (CRSwNP-A) comorbid asthma	58 CRSwNP: 41 (CRSwNP+A: 20 CRSwNP-A: 21) HC *: 17	Belgium	CRSwNP+A: 45.8 CRSwNP-A: 47.5	M: 31 (22) F: 27 (19)	AS: (20) AT: (20) AaS: (7)	MMSS and tissue samples collected during ESS	MT (sequencing, OTUs)	HCs and CRSwNP patients had about the same total bacterial load, but the bacterial diversity and evenness were significantly lower in the CRSwNP group, especially the CRSwNP-A group (both evenness and Shannon’s diversity), compared with HCs. The phylum *Proteobacteria* and genus *Haemophilus* (*H. influenzae*) were more abundant than in HCs. In contrast, *Staphylococcus xylosus* and *Bifidobacterium longum* were less prevalent and abundant than in the HCs. The most abundant species in CRSwNP-A was *S. aureus*, and *E. coli* was found in high amounts in CRSwNP+A.	No antibiotics or systematic corticosteroids in the previous three months *
[77]	CSS	To characterize the sinonasal fungal communities (mycobiota) in CRS patients and DCs via the fungal ITS2 marker amplicon sequencing	144 CRSwNP: 49 CRSsNP: 50 CRS/CF: 7 DC: 38	New Zealand (Most of the patients were European (111/144, 77.1%))	18–84 CRSwNP: 18–71 (50.0)	M: 72 (29) F: 72 (19)	AS: 58 (35) AsS: 18 (15)	MMSS collected during ESS	MT (sequencing, ZOTUs)	At the phylum level, *Basidiomycota* and *Ascomycota* showed the highest RA. The most abundant fungi were *Malassezia*, followed by *Davidiella*.	Most of the patients had not taken antibiotics (41/49, 83.7%) and corticosteroids (44/49, 89.8%) in the previous month
[107]	RCT	To investigate the safety and preliminary efficacy of Manuka honey (MH) with augmented methylglyoxal (MGO) rinses in recalcitrant CRS	25 CRSwNP: 20 (10 with useful information) CRSsNP: 5	New Zealand	27–86 CRSwNP: 49–69	M: 14 F: 11	AS: 10 AR: 12 AsS: 2	Sinonasal swab samples post-EES	SMT	The most common and abundant bacteria were *S. aureus* and *Pseudomonas*.	NA
[108]	CSS	To describe the sinus microbiota of acute exacerbations in CRSwNP, CRSsNP, and allergic fungal rhinosinusitis (AFRS) patients	143 CRSwNP: 55 CRSsNP: 65 AFRS: 14	USA	(52.7) CRSwNP: (53.2)	M: 75 (33) F: 59 (22)	AS: 56 (38) AL: 66 (33) AsS: 9 (7)	Aspiration samples of purulent secretions from within the MM or previously opened sinus	MT (sequencing, OTUs)	The most common bacteria were *Staphylococcus* spp. (*S. aureus* and *S. epidermidis*), *Pseudomonas* spp., *Haemophilus* spp. *Enterobacter* spp., and *Corynebacterium* spp. *Staphylococcus* spp., *Pseudomonas* spp., and *Streptococcus* spp. showed the most RA. An average of 3 taxa per specimen isolated showed a low level of diversity in acute exacerbation CRSwNP	Some patients (no data) had taken antibiotics and corticosteroids
[69]	CS	To demonstrate differential expression of trefoil family factor (TFF) protein genes in CRSwNP patients and the impact of bacterial colonization on their expression.	54 CRSwNP: 29 DC: 25	Croatia	21–69 CRSwNP: 26–69 (53.4)	M: 30 (16) F: 24 (13)	AERD: (3)	Nasal and sinus swab samples collected during FESS	SMT	Most of the patients (23/29, 79.3%) showed positive culture results, of which 15 had isolated pathogenic bacteria and 8 nonpathogenic bacteria (*S. epidermidis*). * Pathogenic bacteria isolated: *S. aureus*, *E. coli*, Group B *S. haemolyticus*, *Morganella morganii*, *Enterobacter* spp., *Serratia marcescens*, *P. mirabilis*, *Enterobacter freundii*, and *K. oxytoca.*	Patients had taken corticosteroids in the previous three weeks and some cases had also taken antibiotics (no data)
[109]	CCS	To identify a microbiome profile in CRSwNP and CRSsNP patients	20 CRSwNP: 10 CRSsNP: 10	Indonesia	18–>50	M: 9 F: 11	NA	Nasal tissue sample collected during FESS	MT (sequencing, OTUs)	The most common bacteria (phyla) were *Proteobacteria*, *Firmicutes*, *Cyanobacteria*, *Fusobacteria*, *Actinobacteria*, and *Bacteoidetes.*	NA
[57]	CS	To identify trends in bacteria isolated and their antibiotic resistance from Korean adults with CRS	510 CRSwNP: 376 (ECRSwNP: 36 NECRSwNP: 22) CRSsNP: 134	Korea	>18	NA	AS: (33) AT: (51)	Purulent discharge in the maxillary and ethmoid sinuses samples collected during ESS	SMT	73.9% of patients (278/376) showed positive culture results. The most common bacteria were CoNS (*S. epidermidis*), *Streptococcus* spp., *Corynebacterium*, *Propionibacterium* spp., *Haemophilus* spp., *S. aureus*, and *Klebsiella* spp. *S. epidermidis*, *Corynebacterium* spp., and *Enterobacter* spp. were significantly associated with ECRSwNP, and *Haemophilus* spp., *Klebsiella* spp., and *P. aeruginosa* with NECRSwNP.	No antibiotics or systemic corticosteroids for at least the previous month
[110]	CSS	To stratify CRS patients based on their microbial community compositions using a probabilistic modeling approach and the traditional phenotypic approach	31 CRSwNP: 8 CRSsNP: 8 CRS/CF: 7 DC: 8	New Zealand	>18	NA	NA	Ethmoidal sinus biopsy specimens collected during FESS	MT (sequencing, OTUs)	*Staphylococcus*, *Streptococcus*, *Propionibacterium*, and *Corynebacterium* were prevalent in a majority but at low RA. In CRSwNP, *Moraxella* and *Stenotrophomonas* were dominant but showed less *Streptococcus* and *Veillonella* than DC.	No antibiotics or systemic corticosteroids in the previous month
[111]	CSS	To identify reactive allergens of IgE antibodies produced locally in NPs	51 ECRSwNP: 46 (4 finally included) DC: 15	Japan	ECRSwNP: 39–77 (53.3)	M: (2) F: (2)	AS: (3)	Swab and NP samples	SMT	*Moraxella catarrhalis*, *Corynebacterium* spp., and *S. aureus* were identified.	NA
[86]	CSS	To characterize the differences in microbiome profiles between CRSwNP patients and DCs	86 CRSwNP: 59 DC: 27	China	CRSwNP: (46.4)	M: 53 (35) F: 33 (24)	AS: 12 (11) AR: 16 (14)	MMSS collected during FESS	MT (sequencing, OTUs)	The predominant bacterial phyla were *Firmicutes*, *Proteobacteria*, *Actinobacteria*, *Bacteroidetes*, and *Fusobacteria*. The predominant bacterial genera were *Lactobacillus*, *Corynebacterium*, *Staphylococcus*, *Streptococcus*, *Erysipelotrichales*, *Escherichia*–*Shigella*, *Haemophilus*, *Enterobacter*, *Propionibacterium*, and *Pseudomonas*. CRSwNP had a lower nasal microbiome richness than DC. The RA of *Actinobacteria* (predominantly *Corynebacterium*), and *Dolosigranulum* was significantly lower in CRSwNP than in DC. *Lactobacillus*, *Escherichia*–*Shigella*, *Turicibacter*, *Clostridium*, *Enterococcus*, and *Romboutsia* were positively correlated with the severity of CRSwNP (Lund–Mackay CT score). Smoking status, asthma, or allergic rhinitis did not change the microbiome distribution.	No antibiotics and corticosteroids in the previous month
[112]	CSS	To compare bacterial community composition and absolute abundances of *S. aureus* and *S. epidermidis* between CRS patients and DCs	54 CRSwNP: 18 CRSsNP: 22 DC: 14	New Zealand (Most of the patients were European (44/54, 81.5%))	CRSwNP: (53.2)	M: 31 F: 23	AS: 14 (10)	MMSS	MT (qPCR and sequencing, OTUs)	The most common bacteria were *Corynebacterium*, *Haemophilus*, *Staphylococcus*, and *Dolosigranulum*. CRSwNP had a significantly higher overall bacterial load than DCs.	Most of the patients had taken antibiotics and corticosteroids in the previous year
[58]	CSS	To investigate the expression of lipopolysaccharide (LPS) and its relationship with glucocorticoid receptors (GRs) in CRSwNP	162 CRSwNP: 112 (ECRSwNP: 49 NECRSwNP: 63)	China	CRSwNP: 13–71 (ECRSwNP: 19–65 (46.0) NECRSwNP: 13–71 (44.0))	M: 91 (65, ECRSwNP: 28, NECRSwNP: 37) F: 71 (47, ECRSwNP: 21, NECRSwNP: 26)	AS: (8, ECRSwNP: 5, NECRSwNP: 3) AT: (5, ECRSwNP: 4, NECRSwNP: 1)	Swab samples and specimens pre-ESS	SMT	82.1% of patients (82.1%) showed positive culture. The positive rate of bacterial culture of different groups was not different. The main bacterial strains were *S. epidermidis*, CoNS, *E. coli*, *S. pneumoniae*, and *K. pneumoniae.*	No corticosteroids in the previous two weeks
[113]	CS	To verify if topical administration of *Lactobacillus lactis* W136 to the nasal and sinus cavities would be safe for CRS patients refractory to medical and surgical treatment	27 (24 completed the study) CRSwNP: 17 CRSsNP: 7	USA	(54.9)	M: 11 F: 13	AS: 18 AL: 5	Nasal swab and brushing samples pre-ESS and post-ESS	SMT and MT (sequencing, OTUs)	Conventional culture: Oropharyngeal flora, CoNS, *S. aureus*, and *P. aeruginosa* were the most common bacteria.	No antibiotics and corticosteroids in the previous month
[87]	CSS	To investigate the prevalence, diversity, and abundance of archaea in the human sinuses and any associations with disease state	60 CRSwNP: 16 CRSsNP: 15 DC: 9 HC: 20	New Zealand (Most of the subjects were European (44/60, 73.3%))	CRSwNP: (52.8)	M: 37 (14) F: 23 (2)	AS: 19 (6)	MMSS collected during ESS	MT (sequencing, ASVs; and digital PCR)	Phyla *Euryarchaeota* and *Thaumarchaeota* were detected. The most abundant bacteria were *Corynebacterium*, *Staphylococcus*, *Moraxella*, *Lawsonella*, and *Haemophilus*. CRSwNP subjects showed significantly decreased alpha diversity than HCs.	Most of the patients (13/16, 81.3%) had not taken antibiotics in the previous month
[114]	CSS	To establish associations among medication usage, the sinus microbiota, and patients’ clinical outcomes	236 CRSwNP: 79 CRSsNP: 77 DC: 45 HC: 35	New Zealand (Most of the subjects were European (191/236, 80.9%))	18–82 CRSwNP: 18–75 (46.0)	M: 100 (26) F: 136 (53)	AS: 78 (72)	MMSS and tissue samples intra-operatively collected	MT (sequencing, ASVs)	The most common bacteria were *Corynebacterium* and *Staphylococcus* The number of observed ASVs was significantly lower when compared to HCs.	Most of the patients (67/78, 85.9%) had not taken antibiotics in the previous month.
[115]	CSS	To analyze the bacterial flora of the nose and paranasal sinuses in CRS patients who underwent ESS over 65 years of age compared to a younger group of patients (<40 years)	529 (269 completed the study) CRSwNP: 150 CRSsNP: 119	Poland	>18	M: 147 F: 122	AS, AL, and AERD	MMSS collected during ESS	SMT	The most common bacteria was *S. aureus*. There were no statistically significant differences between the occurrence of a particular type of bacteria and the presence of NP in both age groups.	No antibiotics and systemic corticosteroids in the previous month
[59]	CSS	To investigate whether the sinus microbiota in CRSwNP is associated with eosinophilic inflammation	37 CRSwNP: 31 (ECRSwNP: 21 NECRSwNP: 10) DC: 6	Republic of Korea	ECRSwNP: (50.6) NECRSwNP: (37.3)	NA	AT: (13, ECRSwNP: 10, NECRSwNP: 3) AS: (8, ECRSwNP: 8) * Patients with AERD were excluded	MMSS	MT (sequencing, OTUs)	The most common bacteria were *Firmicutes* (mainly Staphylococci), *Actinobacteria* (mainly *Corynebacterium*, *Bifidobacterium*, and *Propionibacterium* species), and *Proteobacteria* (mainly *Moraxella*, *Pseudomonas*, *Enterobacter* and *Aggregatibacter*). ECRSwNP: RA of *Anaerococcus*, *Tepidimonas*, and *Mesorhizobium* were significantly decreased and *Lachnoclostridium* increased compared to those in DCs. ECRSwNP patients had higher asthma incidence and clinical severity scores. NECRSwNP: RA of *Lachnospiraceae* was increased compared with that in DCs. *Deinococcus*, *Sphingomonas*, and *Lactobacillus* were positively correlated with serum extracellular vesicles (EVs).	No antibiotics and systemic corticosteroids for at least the previous month
[55]	CSS	To characterize the normal microbiome, assess for any geographical or clinical influences, and identify any changes associated with CRS within and across geographical sites	532 (410 reached the final stage of analysis) CRSwNP: 172 CRSsNP: 99 DC: 139	Australia New Zealand Thailand India Brazil Chile The Netherlands Canada USA	20–75	NA	AS and AsS	MMSS collected during ESS	MT (sequencing, ASVs)	The most abundant bacteria were *Corynebacterium*, *Staphylococcus*, *Streptococcus*, *Haemophilus*, and *Moraxella*. *Corynebacterium* was significantly reduced and *Streptococcus* increased compared to DCs.	Clinicians were free to treat patients
[98]	CSS	To compare the bacterial flora in CRSwNP and CRSsNP patients and investigate a possible link between the type of bacterial flora and the coexistence	470 (458 completed the study) CRSwNP: 245 CRSNsNP: 213	Poland	M: (50.6) W: (49.8)	M: 248 F: 222	AS: 104 (83) AL: 87 (52) AsS and other non-steroidal anti-inflammatory drugs: 53 (44)	MMSS collected during ESS	SMT	Gram-negative intestinal bacilli *Enterobacteriaceae*, CoNS and streptococci, and *S. aureus* were the most common bacteria. No statistically significant relationship was found between bacterial flora and the presence of asthma, hypersensitivity to drugs, or allergy. No statistical significance between the occurrence of a particular flora and the multiplicity of operations.	No antibiotics in the previous month
[88]	CSS	To characterize the microbiome dysbiosis in AERD patients	37 AERD: 17 HC: 17	USA	NA	NA	AERD: (17)	Inferior turbinate swab samples	MT (sequencing, OTUs)	AERD subjects showed reduced bacterial diversity (fewer species per sample and lower Shannon Diversity indexes). *Moraxella*, *Corynebacterium*, *Pseudomonas*, *Staphylococcus*, *Sphingomonas*, *Streptococcus*, *Propionibacterium*, and *Eikenella* showed the highest RA. Overabundance of *Eikenella corrodens*, *M. catarrhalis*, and *Pseudomonas* (*Proteobacteria* phylum) and underabundance of *Corynebacterium* in AERD patients compared to HCs.	No recent use of antibiotics and corticosteroids
[89]	CSS	To analyze the effects of antibiotics on the nasal microbiome and secreted proteome in CRS patients	99 CRSwNP: 40 CRSsNP: 30 DC: 29	Republic of Korea	CRSwNP: (48.8)	M: 67 (27) W 32 (13)	AS: 4 (2) AT 31 (13) AR: 33 (8)	Nasal secretion samples from MM	MT (sequencing, OTUs)	*Corynebacterium* (*Actinobacteria*), *Staphylococcaceae* (*Firmicutes*), *Streptococcaceae* (*Firmicutes*), *Burkholderiaceae* (*Proteobacteria*), *Lachnospiraceae* (*Firmicutes*), *Veillonellaceae* (*Firmicutes*), *Propionibacteriaceae* (*Actinobacteria*), and *Moraxellacea* (*Proteobacteria*) showed the highest level of RA. Shannon and Simpson indexes were significantly decreased across CRSwNP to DCs. The sinonasal microbiota of the CRSwNP showed significantly decreased bacterial diversity. *Firmicutes* and *Bacteroidetes* (*Prevotellaceae*) were significantly decreased compared to DCs. *Cyanobacteria*, *Staphylococcaceae*, *Propionibacteriaceae*, and *Moraxellaceae* were significantly increased compared to DCs. In the NABX (subjects who had not taken antibiotics three months before sampling) group, the Shannon and Simpson indexes were significantly decreased compared to DCs. Shannon and Simpson indexes were significantly lower in the ABX group than in the NABX group. *Streptococcaceae*, *Lachnospiraceae*, and *Neisseriaceae* were significantly decreased in the ABX group compared to the levels in the NABX group.	Some patients (24/40, 60%) had not taken antibiotics in the previous three months
[60]	CSS	To evaluate the bacterial community composition on distinct types of CRSwNP patients (ECRSwNP and NECRSwNP)	73 CRSwNP: 34 (ECRSwNP: 16 NECRSwNP: 18) HC *: 39	China	ECRsWNP: (48.3) NECRSwNP: (28.5)	M: 36 (18, ECRSwNP: 11, NECRSwNP: 7) F: 37 (16, ECRSwNP: 5, NECRSwNP: 11)	AS: (2, ECRSwNP: 2) * Patients with AERD were excluded	MMSS collected during FESS	MT (sequencing, OTUs)	The most common bacteria were *Firmicutes*, *Actinobacteria*, *Proteobacteria*, and *Bacteroidetes*. The most abundant genera were *Staphylococcus*, *Corynebacterium*, and *Dolosigranulum*. The diversity of sinus microbiota (Chao1 and Shannon indexes) was significantly lower in the CRSwNP than in DC. ECRSwNP: *Firmicutes*, *Actinobacteria*, and *Proteobacteria*. *Staphylococcus*, *Corynebacterium*, and *Moraxella.* NECRSwNP: *Firmicutes*, *Actinobacteria* and *Bacteroidetes*. *Staphylococcus*, *Dolosigranulum*, and *Corynebacterium*. *Staphylococcus* was significantly lower in the ECRSwNP compared to HC. The Shannon index significantly decreased only in the NECRSwNP, but not in the ECRSwNP, compared to HCs. *Staphylococcus* (*S. aureus)* abundance was the lowest in the ECRSwNP. The abundance of *S. aureus* was the highest in the NECRSwNP. The abundance of *Moraxella* was significantly decreased in the NECRSwNP compared with that in the ECRSwNP. The abundance of *Haemophilus* was significantly increased in the NECRSwNP compared to HC.	No antibiotics or corticosteroids in the previous month
[64]	CS	To explore nasal microbial diversity effects on the pathogenesis and prognosis of CRSwNP	147 CRSwNP: 77 (NP recurrent: 12, NP non-recurrent: 65) CRSsNP: 36 DC: 34	China	CRSwNP: (46.4) (NP recurrent: (48.6), NP non-recurrent: (49.7))	M: 86 (43, NP recurrent: 5, NP non-recurrent: 36) F: 61 (34, NP recurrent: 7, NP non-recurrent: 29)	AS: 14 (11, NP recurrent: 3, NP non-recurrent: 8) AR: 22 (14, NP recurrent: 4, NP non-recurrent: 10)	MMSS collected during ESS and MM secretions after 1-year post-ESS	MT (sequencing, OTUs)	The most abundant bacteria were *Lactobacillus*, *Corynebacterium*, *Staphylococcus*, *Streptococcus*, *Escherichia*–*Shigella*, *Enterobacter*, *Haemophilus*, *Moraxella*, and *Propionibacterium*. The RA of *Actinobacteria* (*Corynebacterium*), *Chlamydia*, and *Dolosigranulum* was significantly lower than that in DCs. *Lactobacillus*, *Escherichia*–*Shigella*, *Turicibacter*, *Clostridium*, *Enterococcus*, and *Romboutsia* were positively correlated with the severity of CRSwNP (Lund–Mackay CT score). Smoking status, asthma, or allergic rhinitis did not change the microbiome distribution. *Faecalibaculum* had a significant negative correlation with the TNSS of patients with CRSwNP. The abundance of *Actinobacteria* after surgery was significantly higher than before in the NP non-recurrent group, while there was no significant change in nasal flora in the NP recurrent group.	No antibiotics or corticosteroids in the previous month
[94]	CSS	To assess the microbial composition in CRS patients, comparing different sampling methods and disease subtypes	22 CRSwNP: 8 CRSsNP: 6 DC: 8	Republic of Korea	21–76 CRSwNP: (48)	M: 12 (3) F: 10 (5)	AS: 2 (2) AT: 6 (1)	MMSS, tissue biopsied from the uncinate process (UT) and NP tissue collected during ESS	MT (sequencing, 16S rRNA)	*Bacteroidetes*, *Firmicutes*, *Proteobacteria*, *Actinobacteria*, and *Fusobacteria* were the dominant phyla. The Shannon index was significantly decreased in NP compared to UT. *Firmicutes* was remarkably reduced in NP, whereas *Proteobacteria* was more abundant in NP than in UT. *Sphingomonas* and *Sediminibacterium* were enriched in the NP, while *Ralstonia* was reduced in NP. *Prevotella* was significantly and inversely correlated with disease severity.	No antibiotics and systemic corticosteroids in the previous month
[61]	CSS	To investigate the changes in the clinical, histopathological, and hematological properties and the gut and airway microbiota in CRSwNP endotypes	58 CRSwNP: 46 (ECRSwNP: 32 NECRSwNP: 14) HC: 12	Sudan	CRSwNP: (34.7) (ECRSwNP: 34.8 NECRSwNP: (34.7))	M: 26 (20, ECRSwNP: 13, NECRSwNP: 7) F: 32 (26, ECRSwNP: 19, NECRSwNP: 7)	AS: (4, ECRSwNP:4) AL: (1, ECRSwNP: 1)	MMSS (17) and fecal samples (10)	MT (sequencing, OTUs)	In the airway: Reduced alpha diversity in comparison to HCs. *Moraxella*, *Parvimonas*, and *Porphyromonas* increased more in the ECRSwNP than in the NECRSwNP. These bacteria were positively correlated with CT scores and severe disease. *Prevotella* 9, *Succinivibrio*, *Lawsonella*, and *Exiguobacterium* significantly decreased in ECRSwNP. These bacteria were negatively associated with CT scores and endoscopic score eosinophil percentage. In the gut microbiome: *Actinobacteria* phylum and its major genus (*Bifidobacteria*) were remarkably reduced in CRSwNP, mainly in ECRSwNP. *Bifidobacterium* and *Barnesiell* were negatively associated with CT score and endoscopic score. The abundance of Enterobacterales; *Enterobacteriaceae;* and several genera, such as *Enterobacter*, increased in NECRSwNP.	No antibiotics or systemic corticosteroids in the previous month
[78]	CS	To undertake a comprehensive multi-omics assessment of NP tissue transcriptome, proteome, and associated bacterial and fungal microbiome in CRSwNP patients	3 CRSwNP: 3	New Zealand (all patients were of European ancestry)	CRSwNP: 46–59	M: (3) F: (0)	AS: (1)	NP tissue	MT (sequencing, ZOTUs)	The most abundant bacterial genera were *Staphylococcus*, *Corynebacterium*, *Dolosigranulum*, *Anaerococcus*, and *Propionibacterium*. The most abundant fungal genera were *Malassezia*; *Candida*; *Rhodotorula*; and unclassified members of *Malasseziales*, *Dothideomycetes*, *Mycosphaerellaceae*, and *Phaeophaeriacea*.	No antibiotics and corticosteroids in the previous month
[65]	CS	To explore the effects of nasal microbial diversity and inflammatory types on the prognosis of NPs	77 and DC CRSwNP: 77 (NP recurrent: 12, NP non-recurrent: 62)	China	CRSwNP: (NP recurrent (48.6), NP non-recurrent (49.7))	M: (34, NP recurrent: 5, NP non-recurrent: 29) F: (43, NP recurrent: 7, NP non-recurrent: 36)	AS: (11, NP recurrent: 3, NP non-recurrent: 8) AR: (14, NP recurrent: 4 and NP non-recurrent: 10)	MMSS and NP tissue and MM secretions after 1-year post-ESS	MT (sequencing, OTUs)	The most abundant bacteria were *Firmicutes*, *Proteobacteria*, *Actinobacteria*, *Bacteroidetes*, *Fusobacteria*, and *Spirochaetae*. *Actinobacteria*, *Corynebacterium*, and *Dolosigranulum* were significantly lower than in DCs. There was no significant difference in nasal microbiome richness between NP recurrent and non-recurrent groups. At the genus level, the dominant bacteria were *Lactobacillus*, *Staphylococcus*, *Streptococcus*, and *Bacteroides*. *Faecalibaculum* was negatively correlated with the overall nasal symptoms. The RA of *Actinomycetes* and *Corynebacterium* was significantly higher, and Staphylococci was significantly lower, in the NP non-recurrent group than in the NP recurrent group.	No antibiotics and corticosteroids in the previous month
[116]	CSS	To identify and characterize prophages present in *S. aureus* from patients suffering from CRS, concerning CRS disease phenotype and severity	67 CRSwNP: 30 CRSsNP: 28 DC: 9	Australia	NA	NA	NA	Samples collected during ESS	SMT *	*S. aureus* clinical isolates were obtained (in silico: intact prophages encoding human immune evasion cluster genes and more frequent in patients with more severe disease).	NA
[79]	CS	To analyze the alteration in the sinonasal microbiome in CRSwNP and CRSsNP patients after FESS	35 CRSwNP: 20 CRSsNP: 15	India	12–76 (40.0)	M: 19 F: 16	NA	MMSS pre-FESS and post-FESS	SMT	Pre-FESS cultures: MRSA predominantly, followed *by S. aureus*, *Pseudomonas*, *E. coli*, and *Aspergillus*. Post-FESS culture: *S. aureus* and *E. coli.*	Postoperatively patients were prescribed antibiotics
[117]	CSS	To investigate the potential role of *Pantoea dispersa* in rhinosinusitis	390 (274 completed the study)	Taiwan	20–99 (53.6)	M: 156 F: 118	AS: 8 AR: 75	Nasal swab samples	SMT	Seven CRSwNP patients had culture growth of *P. dispersa.*	NA
[40]	CSS	To examine the bacterial communities of the sphenoidal sinus in Iranian patients with and without CRS	36 CRSwNP: 18 DC: 18	Iran	CRSwNP: 30–63 (42.7)	M: 18 (9) F: 18 (8)	AS: 6	Sphenoidal sinus surface mucosa swab samples collected during FESS	MT (qPCR)	The most common bacteria were *Actinobacteria* (*Corynebacterium*) and *Staphylococcus* spp. *S. pneumoniae* and *H. influenza* were not detected in any of the samples. *S. haemolyticus* and *P. aeruginosa* were significantly more prevalent than DCs.	No antibiotics in the previous month
[92]	CS	To determine whether altered nasal microbiota constituents could be used as biomarkers to predict CRSwNP recurrence	85 (60 with complete clinical information for the establishment of a predictive model of CRSwNP recurrence) CRSwNP: 85 (NP recurrent: 39, NP non-recurrent: 46)	China	CRSwNP: 18–73 (46.4) (NP recurrent: (46.2), NP non-recurrent (46.5)	M: (64, recurrent: 28, non-recurrent: 36) F: (21, NP recurrent: 11, NP non-recurrent: 10)	AS: (17, NP recurrent: 15, NP non-recurrent: 2)	MMSS collected during ESS *	MT (sequencing, OTUs)	There was no significant difference in community diversity (OTUs, Shannon diversity index, and Chao richness), but both groups (recurrence and non-recurrence) showed distinct composition. Genera from the *Proteobacteria* and *Firmicutes* phyla were the major taxa that differed in abundance between both groups. *Campylobacter*, *Bdellovibrio*, and *Aggregatibacter* were more abundant than in the recurrence group. *Actinobacillus*, *Gemella*, and *Moraxella* were more abundant in non-recurrence. *Shewanella* and *Preptostreptococcus* exhibited a decrease in abundance, and *Friedmanniella*, *Curvibacter*, and *Pseudoxanthomonas* were more abundant in recurrence than non-recurrence. *Porphyromonas*, *Bacteroides*, *Moryella*, *Aggregatibacter*, *Butyrivibrio*, *Shewanella*, *Pseudoxanthomonas*, *Friedmanniella*, *Limnobacter*, and *Curvibacter* were the most important taxa discriminating recurrence from non-recurrence specimens.	No antibiotics or corticosteroids in the previous month
[118]	CSS	To explore the difference between sinus bacteriology in CRSwNP and CRSsNP patients and to analyze the differences in culture results from swabs taken from the MM versus the ethmoid sinus	448 CRSwNP: 160 CRSsNP: 288	Jordan	(40.0) CRSwNP: (39.7)	M: (96) F: (64)	NA	Ethmoid sinus and MMSS collected during FESS	SMT	The most common bacteria were MRSA, followed by CoNS and *S. aureus*.	No antibiotics or corticosteroids before surgery
[119]	CSS	To investigate the changes in microbiota and cytokines levels in CRSwNP and CRSsNP patients.	36 CRSwNP: 12 CRSsNP: 10 DC: 15	China	CRSwNP: (47.3)	M: 24 (9) F: (13/3)	NA	Secretions collected from the middle nasal canal, maxillary sinus, and ethmoid sinus intra-operatively collected	MT (sequencing, OTUs)	The most common bacteria were *Staphylococcus*, *Corynebacterium*, *Porphyromonas*, *Serratia*, *Pseudomonas*, *Fusobacterium*, *Carnobacterium*, *Dolosigranulim*, *Cultibacterium* (*formerly Propionibacterium acnes)*, and *Lawsonella*. Beta diversity was significantly different between CRSwNP and DC. The abundance of *C. propinquum* and *Carnobacterium maltaromaticum* in CRSwNP differed from that in DC. *Lawsonella*, *Moraxella*, *Corynebacterium*, *Carnobacterium*, and *Hafnia*–*Obesumbacterium* were different at the genus level.	No corticosteroids in the previous month
[82]	CSS	To examine evidence of microbial exposure in subjects by probing serum samples of CRS patients and controls for seroreactivity to microbial protein-directed IgG and IgA	118 CRS: 39 DC: 79	USA	NA	NA	NA	Serum samples	MT (CRS-focused Nucleic Acid Programmable Protein Array (NAPPA))	CRSwNP patients showed elevated sero-reactivity against *S. aureus*. Influenza A virus (H1N1 and H3N2) and rhinovirus B14 were identified.	No antibiotics and corticosteroids in the previous month
[91]	CSS	To demonstrate the role of bacteria in the pathogenesis of fungal ball (FB) versus CRSwNP and investigate the differences in microbiome profiles through a comparative analysis of microbial community diversity	42 CRSwNP: 10 FB: 49 DC: 4	China	CRSwNP: (41.3)	M: 15 (7) F: 28 (3))	NA	MM and superior meatus swab samples	MT (sequencing, OTUs)	The major abundant phyla were *Firmicutes*, *Proteobacteria*, *Bacteroidetes*, and *Actinobacteria*. The abundance of *TM7* (*Saccharimonadia*), *Chloroflexi*, and *Bacteroidete* significantly differed between the CRSwNP and DC groups. The RA of *Ruminococcacea* from the phylum of clostridia and *Comamonadaceae* from the phylum of Burkholderiales was significantly higher, while that of *Lactobacillus*, *Bacteroides* S24-7, and *Desulfovibrio* was significantly lower than DC. The RA of *Haemophilus* was increased in CRSwNP.	No antibiotics in the previous month
[62]	CSS	To compare the nasal bacteriology between ECRSwNP and nECRSwNP patients	CRSwNP: 295	Taiwan	CRSwNP: 20–84 (46.1)	M: (205) F: (90)	AS: (14, ECRSwNP: 9, NECRSwNP: 5) AR: (124, ECRSwNP: 70, NECRSwNP: 54) AsS: (1, NECRSwNP: 1)	MMSS pre-FESS	SMT	The most common bacteria were *S. aureus* and CoNS. Culture rates were similar between ECRSwNP and nECRSwNP. Gram-negative aerobes (mainly *H. influenzae* and *C. koseri*) were significantly more isolated from the NECRSwNP than the ECRSwNP.	No antibiotics in the previous week
[84]	CSS	To evaluate the microbial composition in the context of the inflammatory environment in patients suffering from CRSwNP, AERD, and CRSsNP	80 CRSwNP: 20 CRSsNP: 20 AERD: 20 DC: 20	Austria	CRSwNP: (49.4) AERD: (46.8))	M: 50 (CRSwNP: 16, AERD: 12) F: 30 (CRSwNP: 4, AERD: 8)	AS: 42 (CRSwNP: 16, AERD: 20) AL: 32 (CRSwNP: 8, AERD: 8)	Anterior naris swab samples and MMSS	MT (sequencing, ASV)	Corynebacteria and staphylococci showed the highest RA in both CRSwNP and AERD. No alpha and beta diversity difference was observed between the DC, the CRSwNP, and the AERD groups. *Dolosigranulum* was less prevalent, and *Lawsonella* was more prevalent in patients with NP than in DCs. A higher RA of staphylococci in the MM in AERD patients compared to CRSwNP was observed, as well as of *Lawsonella* in patients suffering from CRSwNP in MM and anterior naris compared to DC.	No corticosteroids in the previous two weeks
[63]	CSS	To characterize nasal dysbiosis in a cohort of ECRSwNP patients and compare their nasal microbiomes with those of HCs	88 ECRSwNP: 34 Patients without CRSwNP: 10 HC: 44	China	18–79 ECRSwNP: 18–67 (43.8)	M: 69 (30) F: 19 (4)	AS: 7 AT: 7	MM brush samples	MT (sequencing, ASVs, and OTUs)	ECRSwNP had higher bacterial alpha diversity (Shannon and Chao1 indexes, intra-individual bacterial diversity). The most dominant phyla were *Actinobacteria* and *Firmicutes*. ECRSwNP was defined by increased RA of *Sphingomonas*, *Moraxellaceae*, *Bacteroides*, *Bifidobacterium*, *Ruminococcus*, and *Faecalibacterium*, as well as by decreased abundances of *Ralstonia*, *Propionibacterium*, and *Propionibacter*. *Sphingomonas* was more abundant in ECRSwNP than in HCs. *Parabacteroides*, *Akkermansia*, *Devosia*, *Sutterella*, and *Desulfovibrio* positively correlated with the SNOT-20 score. The abundances of *Dyella*, *Gordonia*, and *Moraxella* were positively correlated with LM (Lund–Mackay) CT scores, whereas the abundances of *Gemmiger*, *Faecalibacterium*, *Anaeroplasma*, *Sutterella*, *Blautia*, *Geobacillus*, *Bifidobacterium*, *Sphingomonas*, *Dorea*, *Roseburia*, and *Ruminococcus* negatively correlated with LM CT scores.	No antibiotics and corticosteroids in the previous week
[83]	RCT	To assess the longitudinal effect of corticosteroid therapy on sinus microbiota in CRSwNP patients	44 CRSwNP: 29 DC: 15	Canada	NA	M: 25 (14) W 18 (14)	AS: 18 (18) AsS: 11 (11)	MMSS	SMT and MT (microbiotyping using MALDI-TOF-MS)	The most prevalent organisms were *Staphylococcus* and *Corynebacterium.* The difference in the number of isolated organisms and the difference between Gram-positive and Gram-negative isolates were not statistically significant between HCs and CRSwNP patients.	No antibiotics and/or corticosteroids for at least the previous month
[120]	CSS	To evaluate the nasal microbiome, NP inflammation mediators, and inflammatory cell infiltration in CRSwNP patients	77 CRSwNP: 77	China	CRSwNP: 46.4	M: (34) F: (43)	AS: (11) AL: (14)	MMSS collected during ESS	MT (sequencing, OTUs)	The most common genus was *Enterobacter.*	No antibiotics and corticosteroids for at least the previous month
[75]	CSS	To investigate the fungal and bacterial microbiome of sinus mucosa in CRSwNP and CRSsNP patients versus HC	92 CRSwNP: 31 CRSsNP: 31 HC: 30	USA	(50.0)	M: 47 F: 45	AS: 32 (20)	Ethmoid tissue and skull base collected during ESS	MT (sequencing, ASVs)	Two patients (2/31, 6.5%) had positive fungal cultures. The mycobiome composition was not significantly different between HC and CRSwNP. *Saccharomycetales* and *Cutaneotrichosporon* were lower among CRSwNP, and *Alternaria* species were higher among CRSwNP. Beta diversity at the ASV and the genus level differed significantly between CRSwNP and HC. CRSwNP had a significantly greater abundance of *Alternaria* and *Ramularia* and a significantly lower abundance of *Cutaneotrichosporon* than HC. CRSwNP had significantly lower alpha diversity compared with HC.	NA

RCT, randomized controlled trial; CCS, case–control study; CS, cohort study; CSS, cross-sectional study; NP, nasal polyp; CRSwNP, chronic rhinosinusitis with nasal polyps; ECRSwNP, eosinophilic CRSwNP; NECRSwNP, non-eosinophilic CRSwNP; CRSsNP, chronic rhinosinusitis without nasal polyps; ACP, antrochoanal polyps; DC, disease control patient; HC, healthy control subject; M, men with CRSwNP; F, females with CRSwNP; AS, subjects with asthma; AT, subjects with atopia; AL, subjects with allergy; AR, subjects with allergic rhinitis; AsS, subjects with aspirin sensitivity; ASA, subjects with acetylsalicylic acid sensitivity; AERD, subjects with aspirin-exacerbated respiratory disease; CF, cystic fibrosis; MM, middle meatus; MMSS, MM Swab samples; ESS, endoscopic sinus surgery; FESS, functional ESS; SMT, microbial identification using standard microbiological techniques; MT, microbial identification using molecular techniques; OTU, operational taxonomic unit; ZOTU, zero-radius out; ASV, amplicon sequence variant; PCR, polymerase chain reaction; qPCR, quantitative polymerase chain reaction; RT-PCR, reverse-transcriptase PCR; RA, relative abundance; MRSA, methicillin-resistant *Staphylococcus aureus*; CT, computed tomography; SNOT: Sinonasal Outcome Test; TNSS, total nasal symptom scores; NA, not applicable. *: Information not fully confirmed.

#### 3.2.2. Relationship between Microbial Dysbiosis and Inflammation in CRSwNP

The most relevant information extracted from the 21 studies that reported results about the relationship between nasal microbiota and inflammation in CRSwNP patients is summarized in Table 2. Information about each study about the study type, objective, sample size, population/country, age, sex, and other comorbidities of the patients included, as well as sample type and methodology used or other observations, is detailed in Table 1, with the only exception of two studies [121,122] (Appendix A). The methodology employed to identify or measure included molecular techniques such as polymerase chain reaction (PCR) assays, immunoassays, histological analysis, or cytometry, among other techniques.

Studies mainly focused on the bacterial microbiota, including only one ex vivo study with information on nasal fungi and viruses and no study on viruses. In detail, enzyme-linked immunosorbent assay (ELISA) showed that *Aspergillus niger* stimulation increased the pro-inflammatory cytokines tumor necrosis factor-alpha (TNF-α), granulocyte-macrophage colony-stimulating factor (GM-CSF), and IL-6. In contrast, *Cladosporium sphaerospermum*, *Alternaria alternata*, and *Penicillium notatum* stimulation reduced TNF-α and IL-6 but induced under a dose-dependent remodeling of transforming growth factor beta 1 (TGF-β1) and basic fibroblast growth factor (bFGF) [121].

Among the most remarkable bacterial results, Gram-negative bacteria were associated with an increase in blood eosinophils and serum IgE levels, just as CRSwNP patients with activity of toll-like receptors (TLRs) [73,97,102]. The peripheral blood levels of the Gram-negative bacterial product lipopolysaccharide (LPS) were also positively correlated with the expression of the glucocorticoid receptor-beta (GR-β) involved in inflammation modulation [58]. Bacterial biofilm was more frequently found in samples from subjects with non-functional taste receptor 2 member 38 (TAS2R38), a protein associated with the innate bacterial defense mechanism of the human upper airway [72]. Associations between specific bacterial groups of the core microbiota of CRSwNP patients and inflammation patterns were also reported. Thus, bacteria taxa commonly found in CRSwNP patients, including *Corynebacterium* or *Staphylococcus*, were associated with peroxisome proliferator-activated receptor gamma (PPAR-y) and retinoic acid-inducible gene I (RIG-I) signaling pathways involved in the eosinophil activation [103], as well as elevated levels of multiple inflammatory cells and cytokines [104]. A higher concentration of bacteria of the *Enterobacteriaceae* family was found in patients with IL-5-positive NPs [64,120] and associated with higher levels of IL-8 [122]. At the genus level, *Enterobacter* was positively correlated with neutrophils in NP [120]. Several studies also showed a positive correlation between *Bacteroides* and tissue eosinophils count [63,120], and *Moraxella* was associated with an increase in T and B lymphocytes and macrophages [110], just as negatively correlated with blood eosinophils [120]. The relative abundance of *Anaerococcus* was negatively correlated with IL-5 levels in tissues and IL-5-producing innate lymphoid cells (ILC2) [59], although this genus was also negatively associated with tissue proteins and positively correlated with collagen proteins (COL6A3 and COL1A2) in other studies [78]. *Streptococcus* positively correlated with immunoglobulin proteins (immunoglobulin kappa constant, IGKC; and immunoglobulin heavy constant gamma 1, IGHG1) [78]. At the species level, it is notable that *Corynebacterium accolens* was negatively correlated with eosinophilic cationic protein (ECP) [24] and Eotaxin-3 [84], while CRSwNP patients with *S. epidermidis* isolated showed an upregulation of trefoil family factor peptide 3 (TFF3), involved in the preservation of epithelial surface integrity, in middle nasal turbinate and NPs [69].

**Table 2 ijms-25-08223-t002:** Summary of the main findings of the included studies focused on the study of the relationship between microbial dysbiosis and inflammation in CRSwNP. Sample type and methodology employed to identify or measure the microbiota and the different types of cells or factors involved in regulating the immune response.

Ref.	Sample Type	Methodology	Results/Conclusion
[73]	MMSS + blood samples	SMT + NA	More Gram-negative aerobic and facultative anaerobic bacteria in the normal subgroup than in the subgroup with increased blood eosinophils.
[100]	MMSS + tissue samples from the uncinated processor posterior part of the inferior nasal turbinate	SMT + histological analysis (hematoxylin and eosin staining)	Differences in the distribution of neutrophils and macrophages in nasal mucosa were significant between smokers with pathogenic bacteria and non-smokers without pathogenic bacteria.
[97]	MMSS + blood samples	SMT + standard hospital protocol to determine serum biomarkers	CRSwNP patients colonized with Gram-negative bacteria have a similar inflammation pattern to those colonized with only Gram-positive bacteria. Higher serum IgE levels were associated with non-*Pseudomonas* Gram-negative bacteria.
[121] (Ex vivo)	NP tissues	ELISA and nasal polyp explant tissue stimulation model	*Aspergillus niger* stimulation increased pro-inflammatory cytokines tumor necrosis alpha (TNF-α), granulocyte-macrophage colony-stimulating factor (GM-CSF), and interleukin- (IL-6), while having little effect on the remodeling cytokines basic fibroblast growth factor (bFGF) and transforming growth factor beta 1 (TGF-b1). *Cladosporium sphaerospermum*, *Alternaria alternata*, and *Penicillium notatum* stimulation reduced pro-inflammatory cytokines TNF-α and IL-6 but induced a dose-dependent increase in remodeling cytokines bFGF and TGF-b1.
[102] (Ex vivo)	Nasal lavage and sino-nasal mucus samples	MT + transfected human embryonic kidney (HEK293) cells	Two patients only showed toll-like receptor 4 (TLR4) activation correlated positively with TNF and with a more elevated bacterial abundance but lower bacterial diversity than non-activated samples. The microbiota of samples with TLR activity was mainly composed of Gram-negative bacteria (Gammaproteobacteria, including the genera *Haemophilus* or *Moraxella*). Samples without TLR activity showed microorganisms commonly implicated in CRSwNP (*Staphylococcus*, *Corynebacterium*, or *Moraxella*).
[103]	Sinus brushing samples	MT + qRT-PCR	Most of the CRSwNP patients cluster into a subgroup typically dominated by *Corynebacterium*. Most of the CRSwNP patients cluster into a subgroup associated with peroxisome proliferator-activated receptor gamma (PPAR-y) and retinoic acid-inducible gene I (RIG-I) signaling pathways and a significant increase in IL-5.
[104]	Tissue biopsies collected from the bulla ethmoidalis	MT + Cytometric Bead Array (CBA) and transfected human embryonic kidney (HEK293) cells	CRSwNP patients were grouped in subject clusters associated with *Staphylococcus*, *Corynebacterium*, *Streptococcus*, and *Propionibacterium*. IL-8, cluster of differentiation (CD68)-positive macrophages, eosinophils, neutrophils, plasma cells, and IL-5 were significantly elevated in CRSwNP patients.
[72]	Sinonasal mucosa samples + saliva or venous blood	SMT and confocal laser scanning microscopy assay + qRT-PCR and genotyping	Bacterial biofilms were more frequently found in samples from subjects with nonfunctional taste receptor 2 member 38 (TAS2R38).
[24]	MMSS + nasal tissue samples	MT + UniCAP systems	IgE and IL-5 were negatively correlated with *Geobacter anodireducens*/*sulfurreducens* and *Pelomonas puraquae*. IL-5 was negatively correlated with *Corynebacterium macginleyi* and eosinophil cationic protein (ECP) with *Corynebacterium accolens*, *Corynebacterium macginleyi*, and *S. pneumoniae*. ECP and IL-5 were positively correlated with *E. coli* (CRSwNP+AS patients). Specific IgE to staphylococcal enterotoxins (SE-IgE) was significantly more frequent in CRSwNP with asthma and correlated with disease severity.
[69]	Nasal and sinus swab samples + tissue samples	SMT + qPCR and immunochemistry	Patients with normal flora (*S. epidermidis*) showed a significant seven-fold upregulation of the trefoil factor 3 (TFF3) gene in middle nasal turbinate and nasal samples compared to the same tissue specimen from patients with sterile swabs.
[110]	Ethmoidal sinus biopsy specimens	MT + histological analysis	CRSwNP were dominated by *Moraxella* and *Stenotrophomonas* but had less *Streptococcus* and *Veillonella* than DCs. T and B lymphocytes and macrophages were significantly elevated in the CRSwNP cohort compared to DCs.
[58]	Swab samples and specimens pre-ESS + blood samples	SMT + limulus amoebocyte lysate (LAL) assay, immunohistochemistry, and immunofluorescence	An abundant lipopolysaccharide (LPS) was found in the peripheral blood, especially for NECRSwNP patients. LPS levels were positively correlated with glucocorticoid receptor-beta (GR-β) expression in CRSwNP. Neutrophils and macrophages were the principal inflammatory cells containing LPS.
[59]	MMSS + NP tissues	MT + flow cytometry and ELISA	The RA of *Lachnoclostridium* showed a positive correlation with IL-5-producing innate lymphoid cells (ILC2s) and IL-5 levels. In contrast, the RA of *Anaerococcus* showed a negative correlation with IL-5-producing ILC2 and IL-5 levels in tissues.
[122]	Middle nasal passage swab samples + blood serum and nasal lavage fluid	SMT + enzyme immunoassay	The level of cytokines in the nasal lavage fluid (IL-8) and blood serum (IL-1β) was correlated with the total number of microorganisms and the concentration of *Enterobacteriaceae* and Staphylococci.
[64]	MMSS and MM secretions + NP tissues	MT + immunoassay	*Enterobacter* in patients with the IL-5-positive NP group was higher than that of the IL-5-negative NP group.
[78]	NP tissues	MT + transcriptome (RNA sequencing) and proteome analysis	*Corynebacterium* and *Anaerococcus* were negatively associated with tissue structural proteins (vimentin (VIM) and histoneHIST1H4A), actin beta (ACTB), and glyceraldehyde 3 phosphate dehydrogenase (GAPDH). *Anaerococcus* positively correlated with collagen proteins COL6A3 and COL1A2. *Streptococcus* positively correlated with immunoglobulin kappa constant (IGKC) and immunoglobulin heavy constant gamma 1 (IGHG1).
[65]	MMSS + NP tissues	MT + immunoassays and immunohistochemical staining	The RA of *Actinomycetes* and *Corynebacterium* was significantly higher, and Staphylococci was significantly lower in the NP non-recurrent group than in the NP recurrent group. The median expression levels of IFN-γ, IL-8, IL-17A, IL-17E, and IL-18 were significantly higher in the NP recurrent group than in the NP non-recurrent group.
[119]	Secretions collected from the middle nasal canal, maxillary sinus, and ethmoid sinus	MT + ELISA	The abundance of *Cultibacterium acnes* was positively related to IL-2 level, while *Staphylococcus caprae* negatively correlated with IFN-γ.
[84]	Anterior naris and MMSS + nasal-secretion samples	MT + electrochemiluminescence technology	Corynebacteria and Staphylococci were the most abundant taxa in all patient groups. *Dolosigranulum* was also less prevalent, and *Lawsonella* was more prevalent in patients with NP than in DCs. AERD patients showed an increased RA of *Staphylococcus* and a decreased RA of *Corynebacterium* compared to other patient groups. Elevated levels of type 2 response-associated cytokines IL-5, IL-9, and Eotaxin-3 in patients with NP. Eotaxin, chemokine CCL17, and IL-6 levels were significantly elevated in CRSwNP patients compared to DCs. *Staphylococcus* and IL-5 showed a moderately strong correlation, while *Corynebacterium accolens* and Eotaxin-3 were negatively correlated.
[63]	MM brush samples + NP	MT + histological analysis	*Sphingomonas*, *Akkermansia*, *Blautia*, *Devosia*, *Desulfovibrio*, *Parabacteroides*, and *Bacteroides* were positively correlated with absolute tissue eosinophil count. The abundances of *Bifidobacterium*, *Blautia*, *Desulfovibrio*, *Gemmiger*, and *Bacteroides* were positively correlated with the percentage of tissue eosinophils.
[120]	MMSS + NP tissue samples + blood samples	MT + immunoassays and hematoxylin-and-eosin staining, immunohistochemical staining, and flow cytometry	*Enterobacter* was significantly higher in the IL-5-positive NP group than in the IL-5-negative NP group. IL-17 and IL-27 were negatively correlated with *Enterobacter* and *Anaerococcus*. IL-8 was negatively correlated with *Enterobacter* and *Staphylococcus*. IL-18 was positively correlated with *Candidatus*–Arthromitus and *Arthrofilaria* and negatively correlated with *Haemophilus*. IL-27 was positively correlated with *Faecalibaculum*. *Arthrofilaria* was positively correlated, and *Moraxella* and *Peptostreptococcus* were negatively correlated with blood neutrophil counts. *Lactobacillus* and *Enterobacter* were positively correlated with the degree of neutrophil infiltration in NP tissue. *Porphyromonas* was negatively correlated with tissue neutrophil infiltration.

NP, nasal polyp; CRSwNP, chronic rhinosinusitis with nasal polyps; ECRSwNP, eosinophilic CRSwNP; NECRSwNP, non-eosinophilic CRSwNP; CRSsNP, chronic rhinosinusitis without nasal polyps; DC, disease control patient; AERD, subjects with aspirin-exacerbated respiratory disease; MM, middle meatus; MMSS, MM swab samples; SMT, microbial identification using standard microbiological techniques; MT, microbial identification using molecular techniques; ELISA, Enzyme-Linked Immunosorbent Assay; PCR, polymerase chain reaction; qPCR, quantitative polymerase chain reaction; qRT-PCR, quantitative reverse-transcriptase PCR; RA, relative abundance; NA, not applicable.

#### 3.2.3. Effect of the Modulation of the Microbiota on the CRSwNP Treatment

According to the PICO criteria, Table 3 summarizes the evidence found in nine studies in the present systematic review of the impact of different treatments on the nasal microbiota in CRSwNP patients. Again, further information for each of the studies included can be found in Table 1 or (Appendix A). The studies mainly evaluated the effect of mixed medical treatments including antibiotics, such as clarithromycin, doxycycline or amoxicillin–clavulanate; nasal saline irrigation; or oral and systemic corticosteroids, such as mometasone furoate or prednisone [70,83,95,96,105], as well as different alternative to the antibiotics: a natural compound (Manuka honey) [107], a probiotic (*Lactococcus lactis W136*) [113], and a bacteriophage mixture (Otofag) [122]. Three studies involved placebo control groups [105,107,122], two studies included disease control patients [83,95], and only one did not include placebo control [113]. In the rest of the studies, a group of patients who had not taken the primary evaluation treatment was used for comparison. The duration of the treatments ranged from seven days to six months.

Regarding the main results obtained, a significant decrease in alpha diversity and specific bacteria was observed in CRSwNP patients who had taken antibiotics [83,89]. Adding clarithromycin to mometasone furoate spray was also associated with significantly eradicating biofilm formation [70]. The total number of bacteria significantly decreased after the administration of intranasal gel with bacteriophages [122]. Topical administration of alive *L. lactis* W126 reduced the abundance of multiple species of *Staphylococcus*, *Peptostreptococcae*, and *Enterobacialles.* It increased the abundance of *Dolosigranulum pigrum* [113], while Manuka honey reduced the standard semiquantitative bacterial culture rate (mainly about *S. aureus*) in half of the CRSwNP patients [107]. The rest of the studies reported no significant changes in the nasal microbiota [95,96,105]. Some clinical benefits in the experimental groups were reported in three studies [96,113,122].

## 4. Discussion

In recent years, the human microbiota has been identified as a contributing factor in the development and maintenance of diverse pathologies and in the modulation of the immune system. This systematic review comprises original studies focused on the characterization of the host microbiota composition and the study of its influence on inflammatory processes in CRSwNP patients. The results of the evaluation of promising strategies aimed to promote changes in microbiota composition to restore a healthier profile were also included. After reviewing the literature published between January 2014 and February 2024, 70 articles that focused mainly on nasal microbiota were finally included. Although compositional or functional alterations in the gut microbiota play a role in upper-airway diseases such as CRSwNP [29,123], only one study focused on the gut microbiota from fecal samples [61]. The fact that nasal microbiota directly interacts with the nasal epithelial barrier, potentially modulating the local immune response and influencing polyp development [77], as well as constituting a potential therapeutic target [92], can explain the almost absence of studies that comprised gut-microbiota results.

All the variables that could potentially impact the final appreciation of the results included in this review were initially considered: study type, sample size, population, country, patient characteristics (age, sex, and comorbidities), type of sample, and methodology used. It is worth noting that most of the studies were identified as cross-sectional, providing data at a single point in time, unlike the longitudinal studies in which the relative impact of host and environmental factors on human microbiota cannot be further clarified. Evidence of the limitations of this kind of study aimed to identify significant microbiota–disease (CRSwNP) links for helpful diagnostic purposes has previously been described, notably when high inter-subject variability is reported [114,124]. Numerous studies showed high inter-patient variability within the CRSwNP cohort and other cohorts evaluated [55,75,77,88,92,94,102,105,114]. The relatively small sample size of most studies (initial sample size less than 100 subjects) may have led to substantial variability between samples [125,126].

In this context, differences in microbiota composition and inflammatory profile can also be explained by geographical factors, including diet; climate; lifestyle habits, such as smoking; antibiotics; corticosteroid-prescribing patterns; or other environmental exposures [55,75,84,104]. By way of example, previous studies have reported a lower isolation rate of *S. aureus* in patients from China than in Caucasian patients, as well as a different pattern of association for IL-5, nasal polyps, and eosinophilia between European and Chinese patients [4,73]. As mentioned above, the highest number of the studies included were from Asia and Europe. However, in agreement with the results obtained with the only multicenter international cohort study included [55], the composition of the nasal core microbiota appears to be preserved across the countries evaluated. In-depth, the nasal microbiota of CRSwNP patients was primarily composed of the bacterial phyla *Firmicutes*, followed by *Proteobacteria*, *Actinobacteria*, and *Bacteroidetes*, corresponding to the most predominant microbial communities that colonize the upper respiratory tract [127].

Regarding remarkable patient characteristics, on the one hand, the number of male subjects enrolled was slightly higher than that of females, obtaining an average gender ratio (females/males) of 0.85 for the general population and 0.75 for CRSwNP patients. Only one study reported a gender-specific difference in CRSwNP patients, obtaining a significantly higher abundance of *Corynebacterium* in men than in females [95]. On the other hand, the age range was relatively large (from 13 to 88 years), although the age means reported were mainly comprised between 40 and 50 years. Changes in the microbial community composition during aging, probably due to the result of immunosenescence [128], can be observed [127,129]. Although the more frequent occurrence of NPs in older patients was confirmed, Leszczyńska et al. [115] found no statistically significant differences between the occurrence of a specific type of bacteria and the presence of NPs in both patients over 65 years of age and younger patients (from 18 to 40 years).

CRSwNP patients showed a high incidence of asthma, among other comorbidities, such as hypersensitivity to drugs or allergy, being, in some cases, significantly higher than other groups evaluated (CRSsNP patients or control subjects) [114]. Despite being crucial factors for the composition of the microbial core [130,131,132], smoking, history of recurrent exacerbations, endoscopic sinus surgery, or even disease severity could not be analyzed because a significant number of studies evaluated did not detail these data. In contrast, the majority of the studies reported antibiotic- and/or corticosteroid-prescribing patterns. The general criterium in the studies was no antibiotic use within one month before sample collection, as one month appears to be enough time to return the microbiota to a steady level [86,133].

Concerning methodology aspects, first, different types of samples from distinct nasal sites were taken at different moments, before, during, or after surgery, with the middle meatus swab samples and tissue biopsies standing out. As previously mentioned, the microbiota analysis may be more influenced by the sampling technique than the anatomical sampling site [90,93,94]. Swabs taken from the middle meatus are the traditional method of sampling because of their ease of collection, and they likely accurately reflect the sinus microbiological profile, as the middle meatus is the drainage area for the anterior ethmoid sinus, the maxillary sinus, and the frontal sinus, as well as the nasal polyps that usually originate from the uncinate process, ethmoid bulla, or middle turbinate [73,74,118]. In the case of the maxillary sinuses, of note, it is involved in heating and humidifying the air, in the development of odontogenic sinusitis, and in different types of regenerative techniques to treat bone atrophies [134]. In contrast, other studies justified the use of tissue specimens because they incorporate bacterial biofilms that grow on the surface and penetrate the mucosal epithelium [94], whereas sampling from the middle meatus may provide inadequate information [132,135]. The risk of possible sample contamination when using swabs in the sampling method has been described [73].

The authors used culture-dependent techniques, focused on traditional direct growth and identification, and culture-independent techniques, focused on sequencing and molecular analyses. Because most of the human microbiota cannot be cultured, it is necessary to use molecular techniques, which can provide more detailed information on microbiota abundance and diversity [40,74,106]. However, these methods for microbiological characterization have limitations, such as the inability to differentiate between viable and non-viable or commensal and inhaled microorganisms, extensive training, and higher costs [83]. In this context, the differences in the bioinformatics analysis could explain the discrepancies observed between the studies based on the sequencing technology (mainly bacterial 16S ribosomal RNA (rRNA) and fungal internal transcribed spacer (ITS) gene sequencing) [114]. Most studies in the present review analyzed sequencing data by clustering sequences into OTUs at over a 97% sequence similarity threshold. Although OTU clustering is standardly accepted for quantifying microbiological diversity [29,136], it can give misleading results, as it is study-specific and, therefore, not comparable between different datasets [114,137]. It was observed that amplicon sequence variants (ASVs), sequence entities that can be comparable across datasets and provide better resolution and accuracy than the OTU method [137], were employed by more recent studies [55,63,75,84,87,114].

### 4.1. Microbiota Composition in CRSwNP

Although discrepancies were also found, likely due to the limitations described above, the dysbiosis hypothesis as a potential critical explanatory factor of the CRSwNP pathogenesis may be supported by the results of different studies included [55,61,86]. The most predominant microbiota alterations observed in CRSwNP patients compared to controls highlight a trend of lower alpha diversity or level of bacterial diversity within individual samples described with the observed richness (number of bacterial taxa mainly obtained using Chao1 index) and evenness or the relative abundance of those taxa (Shannon or Simpson indexes) [138]. Decreased alpha diversity of the nasal and gut microbiota has been previously associated with the development of inflammatory airway diseases, including asthma or chronic obstructive pulmonary disease (COPD) [139,140,141,142,143,144,145]. Host and environmental factors, such as antibiotics, can directly affect the composition of human microbiota, enhancing the prevalence of this kind of disease [105,140,145,146,147,148]. Nevertheless, the lower diversity mentioned in the present review cannot be explained by the use of antibiotics, as most studies reported that no CRSwNP patients received antibiotics for at least the last month before sampling [24,60,61,75,86,87,88,89].

Enrichment of specific bacteria from the phyla *Proteobacteria* in CRSwNP patients compared with controls—precisely, bacteria from the genera *Haemophilus*, *Moraxella*, and *Pseudomonas*—was also reported across multiple studies [24,38,40,88,89,91]. Considered opportunistic pathogens [139,149], these bacteria have been related to lower-respiratory disease control with increased exacerbation risk [139,140,150]. On the one hand, *Haemophilus* and *Moraxella*, belonging to the class *Gammaproteobacteria*, can induce epithelial damage and expression of host inflammatory pathways [150,151]. A greater abundance of *Moraxella* was found in the CRSwNP subgroup characterized by eosinophil-dominant inflammatory cell infiltration in NPs [60,61,63]. On the other hand, a pathogenic role of *Pseudomonas* and other aerobic and facultative Gram-negative roads has been suggested, as they are infrequently identified from middle meatus from healthy subjects [152]. In this context, it has been described that lipopolysaccharide, the main chemical component of the outer layer of the Gram-negative bacteria cell wall, can induce persistent mucosal inflammation and cause tissue remodeling [153].

In contrast, a decrease in the abundance of *Corynebacterium* and *Dolosigranulum* was also observed [55,64,65,84,86,88,90]. Both respiratory commensal bacteria include specific species that may play a protective role in the nasal airways, modulating the innate immune response and providing better resistance to bacterial and viral pathogens [24,55,139,154]. An improvement in the levels of respiratory cytokines and immune cell populations critical for host defense against microbial infection associated with specific strains of these bacteria has been shown [154]. Commensal bacteria can also exclude pathogens both passively, by competing for space and nutrients, and actively, by secreting antimicrobial compounds [58].

The results obtained for bacteria from the phyla *Firmicutes* were controversial [24,40,55,60,89,90]. It is worth noting that, unlike other *Staphylococcus* species, *S. aureus* colony formation in the nasal mucosa tissue can be essential in the sustained inflammatory response of CRS [39,155]. Although multiple authors have suggested that alterations of bacterial community may play a more significant role in the development of CRSsNP than CRSwNP [57,85,102,109,156,157], *S. aureus* is more prevalent in the latter clinical phenotype [39]. The staphylococcal superantigen hypothesis postulates that *S. aureus* secretes relevant enterotoxins, which induce the local inflammatory response, promoting an increase in cytokines and other inflammatory mediators’ synthesis and eosinophil infiltration in epithelial tissue and inducing an increase in IgE levels [39,63,115,155]. The disruption of intercellular adhesion factors and damage to the epithelial ciliary transport system promoting mucosal barrier defects could also be associated with the colonization of intranasal *S. aureus.* Barrier dysfunction can contribute to bacterial invasion and biofilm formation, resulting in an intensification of local inflammation and polyp development [39,155].

The limited number of results obtained concerning nasal fungal and viral communities suggests that these microbial taxa are unlikely to play a direct pathogenic etiological role in CRSwNP. Further research focused on differentiating whether the identified nasal mycobiota might be a local functional community or simply reflect inhaled environmental fungal material, which has been previously indicated [77]. In the case of respiratory viruses, viral respiratory infection may play a minor role in symptom exacerbation in CRSwNP than CRSsNP [81]. Despite that, viral and fungal dysbiosis may exacerbate the inflammatory process and, thus, contribute to CRSwNP symptomatology through an indirect/secondary role, influencing the bacterial microbiota, deteriorating the epithelial barrier, or persisting in the mucosal inflammation [75,77,158].

Regarding the controversy found in the influence of the simultaneous presence of relevant comorbidities on the nasal microbiota, it is notable that multiple studies obtained differences in the nasal microbiota of CRSwNP patients with and without asthma [24,38,97], the most common comorbidity reported. An intimate association between the gut and respiratory microbial communities, host immune development, and the pathogenesis and manifestation of asthma has been previously suggested [159]. Likewise, it is remarkable that there is a higher abundance of *Moraxella* in the CRSwNP subtype, in which tissue eosinophils predominate among inflammatory cells (ECRSwNP) [60,61,63]. A greater risk of NP recurrence, as well as a worse prognosis post-surgery, have been found in patients with tissue eosinophilia [65,158,160]. Evidence suggests that eosinophils contribute to histological tissue remodeling and, thus, NP development by releasing mediators, pro-inflammatory cytokines, and toxic proteins such as ECP [161]. Finally, although different demographics or biological parameters were not found to be statistically significant in causing the recurrence of polyposis in a recent study [162], differences in the abundance of specific bacterial genera between recurrent and non-recurrent groups were reported [64,92].

### 4.2. Relationship between Microbial Dysbiosis and Inflammation in CRSwNP

The present systematic review also revised possible connections between nasal dysbiosis and inflammation markers. Gram-negative bacteria have been one of the most extensively analyzed mediators for inflammatory reactions. In detail, members of this group of bacteria were positively associated with the activity of transmembrane receptors, such as TLRs, with a prominent role in recognizing pathogens’ components, thus stimulating inflammatory factors’ production and immune cell activity [102]. By way of example, the same author found, in two different studies, a clear correlation between *Enterobacter* and patients with NPs positive for IL-5, a major cellular inflammatory factor in eosinophil-mediated Th2-type inflammation [64,120]. It has been suggested that this commonly conditional pathogen in the human gut can trigger the release of IL-5 when colonizing the nasal cavity, leading to a cascade activation and eosinophil recruitment, further aggravating the Th2 inflammatory response [120].

Likewise, different Gram-positive bacterial genera, such as *Staphylococcus*, *Lachnoclostridium*, or *Streptococcus*, were positively associated with eosinophilic inflammation, elevated levels of type 2 response-associated cytokines and immunoglobulin proteins [59,78,84,104]. A negative association with tissue structural proteins was also reported, which could be linked to a loss of mucosal epithelial disruption, a predisposition toward eosinophil infiltration, and increased permeability to diverse microorganisms into submucosa [104]. Thus, an epithelial barrier with compromised integrity can expose more TLRs to pathogens, promoting the initiation of the inflammation response [110]. Tissue-related effects of specific common airborne fungal species, mainly inducing and remodeling pro-inflammatory cytokines, have been shown in ex vivo NP tissue [121].

On the other hand, specific bacterial species with a potential protective function were negatively correlated with cytokines and other mediators of the host defense response, as in the case of *C. accolens* with eotaxin-3 [84] and ECP [24]. The antimicrobial potential of *C. accolens* against *S. aureus* and methicillin-resistant *S. aureus* (MRSA) clinical isolates or *S. pneumoniae* has also been shown [163,164]. Similarly, a higher expression of peptides involved in preserving epithelial surface integrity was associated with *S. epidermidis* [69]. This commensal bacterium, found in great abundance in human skin and mucosa, has a well-documented role in the control and inhibition of respiratory tract pathogens through the secretion of antimicrobial compounds or the stimulation of the host immune response [165].

As previously suggested, the lower bacterial diversity commonly found in CRSwNP could also contribute to the host’s innate and adaptive immune response [102]. The presence of bacterial biofilm, with a detection rate reaching over 44% of the cases, can also be implicated as an essential feature of sinonasal infections in CRSwNP, acting as a reservoir of pathogenic bacteria and conferring high resistance to host antimicrobial defense strategies [29,72].

Remarkably, the geographic and ethnic variations in the CRSwNP pathophysiology, with a predominance of eosinophilic inflammation involving mainly Th2 cytokines in European and American patients and a predominance of non-eosinophilic and mixed Th1, Th2, and Th17 cytokines patients from East Asia, could explain the differences in the immune response and cytokine profiles associated with specific bacterial groups [119,120]. Apart from this, it seems clear that the role of the nasal microbiota in the CRSwNP pathophysiology causes an onset of cell-mediated immune responses, changes in the cytokine cascade, and defects in the epithelial barrier. However, other non-microbial and host factors, such as innate immune deficiencies, anatomic abnormalities, genetic predisposition, or the co-occurrence of comorbidities, must be considered as modulators influencing disease establishment and severity, as previously noted [139].

### 4.3. Effect of the Modulation of the Microbiota on the CRSwNP Treatment

Taking into account the evidence of a relationship between alterations in the microbiota and CRSwNP, different studies have assessed the effect of different treatments and adjuvants commonly employed in CRSwNP therapy, such as antibiotics, corticosteroids, or nasal saline irrigation, as well as other novel alternative strategies, including microbiome-based therapeutics, on the modulation of the dysbiotic nasal microbiota. As previously mentioned [122], the efficacy of these treatments is not primarily dependent on pathogen removal. Instead, their focus is on restoring a healthy microbial balance or eubiosis.

Due to their anti-inflammatory properties, corticosteroids and antibiotics have been recommended as a first-line CRS therapy. In addition to antimicrobial properties, the current literature implies that long-term macrolide and doxycycline treatment can inhibit neutrophilic and eosinophilic inflammation or even reduce polyp size [165]. Considering the results collected in the present review, no significant changes in microbial communities in CRSwNP patients were found before and after nasal saline irrigation or corticosteroid therapies [70,83,95,105]. Different potential benefits in the sinonasal mucosa of both treatment options have been previously proposed, such as the alteration and elimination of antigens, biofilms, and inflammatory mediators, as well as the contribution to a diverse and balanced sinonasal microbiota [166,167].

The combined treatment of clarithromycin with mometasone furoate nasal spray did not cause significant changes in the nasal microbiota in the study carried out by Varvyanskaya and collaborators [96]. In contrast, a significant eradication in biofilm formation was observed in another study, even with a shorter treatment but different initial daily dose [70]. A decrease in the alpha diversity in patients who had taken antibiotics during the three months before sampling compared to those without antibiotic treatment was also reported [89], denoting the importance of considering antibiotic use when evaluating the association between microbiota and a specific disease state. Based on these results and the trend of a lower alpha diversity observed in CRSwNP patients compared to controls, the antibiotics can negatively affect the microbiota and, consequently, the host physiology, increasing the risk of airway inflammation [159].

Even though, the proportion of clarithromycin-resistant strains remained constant throughout the antibiotic therapy in a study previously cited [96], long-term antibiotic treatment, as well as the repeated use of antibiotics in recalcitrant, difficult-to-treat patients, can increase the risk of inducing significant bacterial resistance to this kind of drug. Thus, given the general trend to dispense with antibiotics in situations where there is an apparent lack of infection according to clinical practice guideline recommendations [9,83], the search for novel antimicrobial alternatives to conventional treatments is an increased area of interest. Changes, but non-significant, in the abundance of multiple bacterial species, total number, and culture rate associated with administering this kind of treatment were found in fewer studies [107,113,122], reflecting the need for more research. Regarding probiotics with a therapeutic potential because of their well-known role in decreasing the density of specific microbial pathogens and modulating the immune response [29], it is necessary to note that their use through intranasal irrigation can also become problematic in specific contexts. For example, probiotic strains can represent pathogen agents after surgery, since the anatomical barriers have been removed, or in patients with significant defects in the epithelial barrier with inappropriate immune response [113].

Finally, the fact that treated groups reported improved sinonasal symptoms despite no significant changes in the nasal microbiota supports the hypothesis that these treatments also show non-microbiological mechanisms. Enhancements in the different natural defense mechanisms of the host respiratory tract, including mucociliary clearance, and the reduction in specific inflammatory mediators associated with these treatments likely contribute to their beneficial effects [95].

## 5. Conclusions

The literature related to human microbiota and CRSwNP from the last ten years was evaluated in the present systematic review. Our primary focus was on the microbiota composition found in CRSwNP patients and the significance of microbiota dysbiosis or alterations in the composition and function of the inflammatory processes in CRSwNP. Studies investigating current and promising therapeutic approaches to manipulate the microbiota in CRSwNP were also included.

As discussed above, the main limitation of this review was related to the absence of uniformity in both the populations studied and the methodologies employed. Nevertheless, the results of the included studies suggest that nasal microbiota dysbiosis can be accepted as a critical factor influencing the onset or severity of CRSwNP. Thus, as has been previously reported for other chronic inflammatory diseases, CRSwNP appears to be a complex interplay, involving innate immune deficiencies, anatomic abnormalities, and genetic predisposition, influenced by the alterations in microbiota composition and diversity.

To sum up, this review provides a valuable understanding of the relationship between nasal microbiota and CRSwNP pathophysiology. Although biological therapies targeting inflammatory mediators, such as the monoclonal antibodies dupilumab, omalizumab, or mepolizumab, have currently been found to be effective in CRSwNP patients, approaches aimed at modulating the host microbiota could become an essential part of combined CRSwNP treatment. Indeed, it is important to remark that further studies into humans are clearly needed to understand the contribution of both nasal and gut microbiota in respiratory diseases as the design of potential therapeutic options based on microbiota modulation.

## 6. Future Directions

A high abundance of several pathogen-related taxa and low microbial diversity, associated with changes in the host’s innate and adaptive immune response and defects in the epithelial barrier, seem characteristic of CRSwNP patients. However, the specific bacterial taxa changes and the associated immune response and cytokine profiles were not always consistent across studies. These observations allow for speculation on the future directions for microbiological diagnosis and treatment of CRSwNP. It seems conceivable that future research focused on the geographic and ethnic variations in host microbiota could provide insights into the influence of environmental and intrinsic factors on CRSwNP, thereby identifying commonly relevant microbial targets and region-specific considerations. In this context, further longitudinal studies that provide an overview of microbiota changes over time would help to clarify whether dysbiosis is a cause or effect of the disease and identify key microbial taxa preceding CRSwNP onset. Future studies should also investigate the roles of nasal viruses and fungi and gut microbiota, providing a more comprehensive view of the CRSwNP aetiology. Advanced omics technologies, such as metagenomics, transcriptomics, or metabolomics, could elucidate how gut microbiota influences nasal health and the mechanistic underlying microbiota, immune response, and barrier integrity.

Finally, the identification of specific microbiota profiles as potential diagnostic biomarkers could lead to non-invasive methods for the early detection and monitoring of CRSwNP. Well-designed clinical trials with extensive CRSwNP patient recruitment could enable researchers to develop appropriate and personalized microbiota-modulating strategies adapted to specific microbiota profiles, enhancing treatment efficacy and minimizing adverse effects.

## Figures and Tables

**Figure 1 ijms-25-08223-f001:**
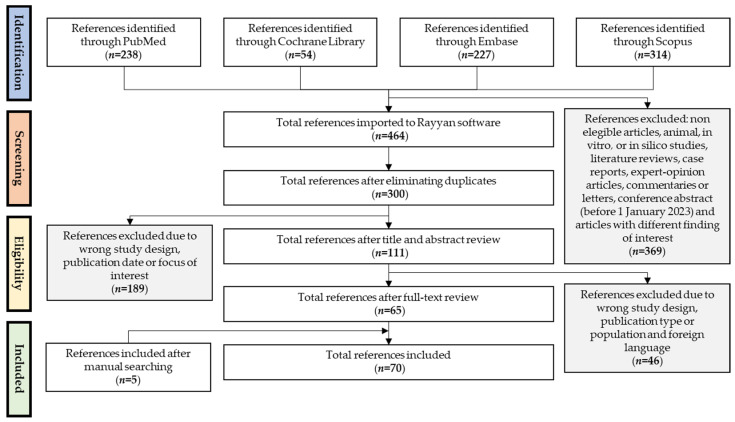
A flow diagram shows the screening and selection process of the studies through the different stages of the present systematic review. *n*: number of references/studies included or excluded.

**Figure 2 ijms-25-08223-f002:**
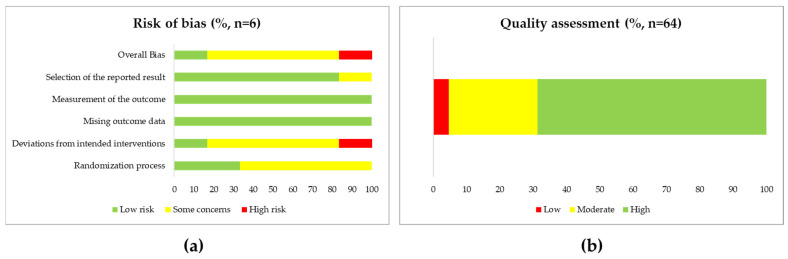
(**a**) Bar graph showing results of risk of bias (RoB) assessment obtained for the randomized controlled trials (*n* = 6) included in this review, using version 2 of the Cochrane risk-of-bias tool (RoB 2) [52]. (**b**) Bar graph showing results of quality assessment obtained for the non-randomized controlled studies (*n* = 64) included in this review, using the Newcastle–Ottawa Scale (NOS) tool [53,54].

**Figure 3 ijms-25-08223-f003:**
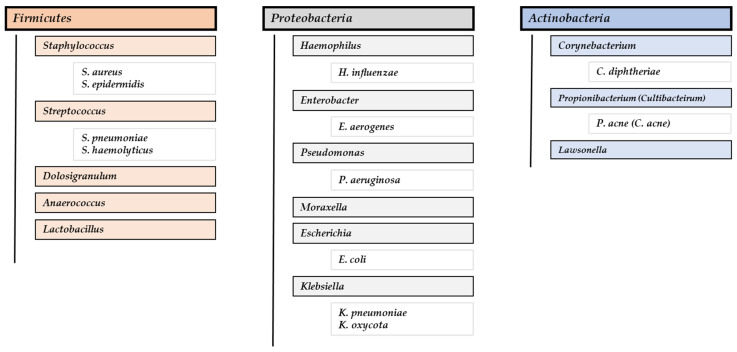
Dominant bacterial phyla, genera, and species found in CRSwNP patients.

**Table 3 ijms-25-08223-t003:** Summary of the main findings from the included studies focused on the evaluation of microbiota modulation in the CRSwNP treatment.

Ref.	Population (P)	Intervention (I)	Comparison (C)	Outcome (O)
[96]	CRSwNP patients underwent bilateral FESS	Clarithromycin + MFNS Duration: Three or six months	MFNS	No significant changes in the nasal microbiota: The proportion of clarithromycin-resistant strains remained constant throughout the antibiotic treatment. *Some clinical benefits were observed in the antibiotic-treated groups: lower CT scores and eosinophil cationic protein (ECP) levels than the control group.*
[70]	CRSwNP patients underwent ESS	Clarithromycin + MFNS Duration: Eight weeks	MFNS	Adding clarithromycin to MFNS was associated with a significant eradication in biofilm formation (six of 12 biofilm-positive samples before surgery turned negative) when compared to MFNS alone (one of 11 biofilm-positive samples). *It was not demonstrated that combined therapy was superior in improving tomography (CT) and symptom (SNOT-20) scores.*
[95]	Post-surgical CRSwNP patients with patent maxillary antrostomy	Treatment of nasal saline irrigation or intranasal corticosteroid spray of any type * CRSwNP patients who used saline irrigation also used topical corticosteroids (spray or irrigation)	DCs	No significant changes in the nasal microbiota associated with saline irrigation or intranasal corticosteroid spray.
[105]	CRSwNP and CRSsNP patients for whom ESS was indicated	Doxycycline or prednisone Duration: Seven days * All patients continued with daily topical corticosteroids nasal sprays and regular sinonasal saline lavage	Placebo	In individual patients, doxycycline was associated with an increase in RA of *Corynebacterium* and *Haemophilus* compared to prednisone and untreated controls.
[107]	CRSwNP and CRSsNP patients underwent ESS	Manuka honey (MH) with augmented methylglyoxal (MXO) sinonasal rinses Duration: Two weeks and ten days of placebo tablets or culture-directed oral antibiotic tablets (control group)	Placebo (saline sinonasal rinses)	50% of patients had a reduction in standard semiquantitative bacterial culture rate (mainly about *S. aureus*) after the MH + MXO treatment. *A statistically significant reduction in LKS (Lund–Kennedy score) outcome was also observed after the treatment.*
[113]	CRSwNP and CRSsNP patients with previous ESS	Topical administration of live *Lactococcus lactis W126* Duration: Two weeks * All patients continued with saline irrigation	No placebo control was included	Treatment was associated with a reduced abundance of multiple species of *Staphylococcus*, *Peptostreptococcae*, and *Enterobacialles*, as well as an increased abundance of *Dolosigranulum pigrum.* *The response pattern showed progressive improvement during the treatment and post-treatment (SNOT-22 scores, nasal congestion, post-nasal drip, and need to blow nose).*
[89]	CRSwNP, CRSnNP patients, and DCs	Oral antibiotics treatment within three months before sampling	No antibiotics treatment	Alpha diversity (Shannon and Simpson indexes) was significantly lower in patients who had taken antibiotics than in patients who had not. *Streptococcaceae*, *Lachnospiraceae*, and *Neisseriaceae* were significantly decreased in patients who had taken antibiotics compared to patients who had not.
[122] *	CRSwNP patients underwent FESS	Intranasal gel with bacteriophage mixture (Otofag) Duration: Ten weeks	Placebo	The total number of bacteria significantly decreased in the experimental group compared to the placebo group. In detail, a significant decrease in the number of *Streptococci* and *Enterobacteriaceae* was observed in the experimental group. The decrease in cytokines in the nasal lavage fluid (IL-8) and blood serum (IL-1B) was proved by the total number of microorganisms and by the concentration of *Enterobacteriaceae* and *Staphylococci*.
[83]	CRSwNP patients	Prednisone + amoxicillin–clavulanate or prednisone alone Duration: Prednisone for three weeks and amoxicillin–clavulanate for two weeks * All patients were concurrent to their baseline intranasal saline irrigation and topical corticosteroid	DCs	Non-significant changes in the nasal microbiota. *Micrococcus* was isolated before both treatments but not post-treatment. After three months from treatment, an increase in the prevalence of *Corynebacterium* was observed in the group treated with prednisone and amoxicillin–clavulanate, as well as in Gram-negative (*Pseudomonas*) and *Staphylococcus* genera in the group treated only with prednisone. *Both treatments showed significant short-term (after one month from treatment) improvement (SNOT-22), but not in the long term (after three months).*

CRSwNP, chronic rhinosinusitis with nasal polyps; CRSsNP, chronic rhinosinusitis without nasal polyps; DC, disease control patient; ESS, endoscopic sinus surgery; FESS, functional ESS; MFNS, mometasone furoate nasal spray; CT, computed tomography; SNOT, Sinonasal Outcome Test. *: Additional information. Reported clinical benefits are shown in italics.

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
