# Peer review of "The Role of the Gut and Airway Microbiota in Chronic Rhinosinusitis with Nasal Polyps: A Systematic Review"

_ijms, 2024, doi:10.3390/ijms25158223_

Round 1
Reviewer 1 Report
Comments and Suggestions for Authors
The aim of this manuscript is to provide a broad overview of the up-to-date understanding of the microbiota composition in CRSwNP patients, the potential mechanism underlying key microbial communities and inflammatory processes in nasal polyps, as well as the current and promising therapeutic strategies involving the modulation of the bacterial microbiota for the CRSwNP treatment.
The study is within the journal’s scope, and I found it to be well-written, providing sufficient information. Anyway, there are some suggestions necessary to make the article complete and fully readable. For these reasons, the manuscript requires major changes.
Please find below an enumerated list of comments on my review of the manuscript:
MINOR POINTS:
If possible, the authors should provide a list of the abbreviations, mentioned in this manuscript.
MAJOR POINTS:
INTRODUCTION:
LINE 45: Chronic rhinosinusitis (CRS) is an inflammatory disorder of the paranasal sinuses and nose that persists for at least 3 months despite getting adequate medical therapy. The term “rhinosinusitis” is currently in use since sinus mucosa irritation and nasal inflammation almost often coexist . CRS is a debilitating disease with symptoms that might be similar to acute cases, nasal blockage, face pressure or fullness, anterior or posterior nasal discharge, olfactory loss, headache, fatigue, anosmia, and cough. These symptoms are typically presented as a result of the multifactorial etiology of the CRS (see, for reference: Abdulrashid NA, Ali OI, Elsharkawy MA. Effect of photobiomodulation therapy on headache, and fatigue in patients with chronic rhinosinusitis: a randomized controlled study. Lasers Med Sci. 2024 Feb 15;39(1):62. doi: 10.1007/s10103-024-04011-4. PMID: 38358423). The authors should provide a complete overview of CRS, according to recent evidence on this topic and highlighting the challenges associated to this disease.
DISCUSSION:
LINE 527: Maxillary sinus plays an important role in heating and humidifying the air, and is involved in different types of regenerative techniques to treat bone atrophies (see, for reference: https://doi.org/10.23812/j.biol.regul.homeost.agents.20233705.232). This manuscript may benefit from providing a brief description of the function of the maxillary sinus in air heating and regenerative processes, and in the development of odontogenic sinusitis: this will be useful to improve the impact and quality of this manuscript, making also the manuscript clear for readers experts and non-experts in the field.
The interesting main topic, described by the authors and the great clinical impact are the most important feature of this manuscript, which is very original and a significant contribute to the ongoing research, due to the extension of the research field on the the up-to-date understanding of the microbiota composition in CRSwNP patients, the potential mechanism underlying key microbial communities and inflammatory processes in nasal polyps, as well as the current and promising therapeutic strategies involving the modulation of the bacterial microbiota for the CRSwNP treatment.
Overall, the contents are rich, and the authors also give their deep insight for some works.
Furthermore, the results are reliable and adequately discussed, since the manuscript relies on a multitude of analysis, to derive its conclusions.
The conclusion of this manuscript is perfectly in line with the main purpose of the paper: the authors have designed and conducted the study properly. As regards the conclusions, they are well written and present an adequate balance between the description of previous findings and the results presented by the authors.
Finally, this manuscript also shows a basic structure, properly divided and looks like very informative on this topic. Furthermore, figures and tables are complete, organized in an organic manner and easy to read.
In conclusion, this manuscript is densely presented and well organized, based on well-synthetized evidence. The authors were lucid in their style of writing, making it easy to read and understand the message, portrayed in the manuscript. Besides, the methodology design was appropriately implemented within the study. However, many of the topics are very concisely covered. This manuscript provided a comprehensive analysis of current knowledge in this field. Moreover, this research has futuristic importance and could be potential for future research. However, major concerns of this manuscript are with the introductive section: for these reasons, I have major comments for this section, for improvement before acceptance for publication. The article is accurate and provides relevant information on the topic and I have some major points to make, that may help to improve the quality of the current manuscript and maximize its scientific impact. I would accept this manuscript if the comments are addressed properly.
Author Response
The aim of this manuscript is to provide a broad overview of the up-to-date understanding of the microbiota composition in CRSwNP patients, the potential mechanism underlying key microbial communities and inflammatory processes in nasal polyps, as well as the current and promising therapeutic strategies involving the modulation of the bacterial microbiota for the CRSwNP treatment.
The study is within the journal’s scope, and I found it to be well-written, providing sufficient information. Anyway, there are some suggestions necessary to make the article complete and fully readable. For these reasons, the manuscript requires major changes.
The authors thank the reviewer for their thorough and meticulous analyses and evaluation of this review paper. Undoubtedly, the comments and suggested changes provided by the reviewer will enhance the final quality of the work.
Please find below an enumerated list of comments on my review of the manuscript:
MINOR POINTS:
If possible, the authors should provide a list of the abbreviations mentioned in this manuscript.
Thank you for pointing this out. A list of the abbreviations mentioned in the main text of this manuscript is now provided (Page 2, Line 44).
MAJOR POINTS:
INTRODUCTION:
LINE 45: Chronic rhinosinusitis (CRS) is an inflammatory disorder of the paranasal sinuses and nose that persists for at least 3 months despite getting adequate medical therapy. The term “rhinosinusitis” is currently in use since sinus mucosa irritation and nasal inflammation almost often coexist . CRS is a debilitating disease with symptoms that might be similar to acute cases, nasal blockage, face pressure or fullness, anterior or posterior nasal discharge, olfactory loss, headache, fatigue, anosmia, and cough. These symptoms are typically presented as a result of the multifactorial etiology of the CRS (see, for reference: Abdulrashid NA, Ali OI, Elsharkawy MA. Effect of photobiomodulation therapy on headache, and fatigue in patients with chronic rhinosinusitis: a randomized controlled study. Lasers Med Sci. 2024 Feb 15;39(1):62. doi: 10.1007/s10103-024-04011-4. PMID: 38358423). The authors should provide a complete overview of CRS, according to recent evidence on this topic and highlighting the challenges associated to this disease.
We appreciate the reviewer’s comment and agree that our manuscript should provide a more complete overview of CRS. According to the reviewer’s recommendation, we have included more information on CRS, highlighting the challenges associated with this disease (Page 3, Lines 49-60).
“Chronic rhinosinusitis (CRS) is characterized by an inflammatory disorder involving the nose, paranasal sinuses, and upper airways, persisting for at least 12 weeks despite appropriate medical therapy. This debilitating disease affects 5-12% of the general population, and common symptoms, which might be similar to acute cases, include anterior or posterior nasal discharge/congestion, facial pain or pressure, impairment of smell or anosmia, and difficulty breathing through the nose, headache, fatigue, and cough [2,3]. Anatomical changes of the nose and paranasal sinuses contribute to the presence and recurrence of CRS symptoms. Imaging studies have shown a significant reduction in the size of the maxillary, sphenoid, and frontal sinuses, along with near-total opacification of these areas and the ethmoid air cells. There is also a rarefaction of the ethmoid bony septae, as well as the ostiomeatal units, sphenoethmoidal recesses, and frontal outflow tracts, are compromised due to thickened mucosa and retained secretions [4–7]”.
DISCUSSION:
LINE 527: Maxillary sinus plays an important role in heating and humidifying the air, and is involved in different types of regenerative techniques to treat bone atrophies (see, for reference: https://doi.org/10.23812/j.biol.regul.homeost.agents.20233705.232). This manuscript may benefit from providing a brief description of the function of the maxillary sinus in air heating and regenerative processes, and in the development of odontogenic sinusitis: this will be useful to improve the impact and quality of this manuscript, making also the manuscript clear for readers experts and non-experts in the field.
Again, we have included relevant information about the maxillary sinus following the reviewer’s recommendation (Page 50, Lines 605-607).
“In the case of the maxillary sinus, of noting that it is involved in heating and humidifying the air, in the development of odontogenic sinusitis, and in different types of regenerative techniques to treat bone atrophies [136]”.
The interesting main topic, described by the authors and the great clinical impact are the most important feature of this manuscript, which is very original and a significant contribute to the ongoing research, due to the extension of the research field on the the up-to-date understanding of the microbiota composition in CRSwNP patients, the potential mechanism underlying key microbial communities and inflammatory processes in nasal polyps, as well as the current and promising therapeutic strategies involving the modulation of the bacterial microbiota for the CRSwNP treatment.
Overall, the contents are rich, and the authors also give their deep insight for some works.
Furthermore, the results are reliable and adequately discussed, since the manuscript relies on a multitude of analysis, to derive its conclusions.
The conclusion of this manuscript is perfectly in line with the main purpose of the paper: the authors have designed and conducted the study properly. As regards the conclusions, they are well written and present an adequate balance between the description of previous findings and the results presented by the authors.
Finally, this manuscript also shows a basic structure, properly divided and looks like very informative on this topic. Furthermore, figures and tables are complete, organized in an organic manner and easy to read.
In conclusion, this manuscript is densely presented and well organized, based on well-synthetized evidence. The authors were lucid in their style of writing, making it easy to read and understand the message, portrayed in the manuscript. Besides, the methodology design was appropriately implemented within the study. However, many of the topics are very concisely covered. This manuscript provided a comprehensive analysis of current knowledge in this field. Moreover, this research has futuristic importance and could be potential for future research. However, major concerns of this manuscript are with the introductive section: for these reasons, I have major comments for this section, for improvement before acceptance for publication. The article is accurate and provides relevant information on the topic and I have some major points to make, that may help to improve the quality of the current manuscript and maximize its scientific impact. I would accept this manuscript if the comments are addressed properly.
Again, the authors are grateful to the reviewer for these comments.
Please see the attachment.

Reviewer 2 Report
Comments and Suggestions for Authors
Please consider the following comments and suggestions in your revision.
Overall, the review paper is well written and it’s a comprehensive review which includes important studies in the field. However, it should the following points need to be incorporated.
1/ Abstract is very generic, and it doesn’t accurately reflect the entire contents in the review paper. Authors should include reflections why this review is important and major works should be highlighted.
2/ Please avoid abbreviations from the key word. And delete the phrase ‘Keywords (List three to ten). It is copied.
3/ In the introduction section, what does it mean ‘The review includes both adult and children’s studies and excludes literature reviews? Why are literature reviews excluded from the study?
3/ Some phrases are not written appropriately such as ‘Three review authors carried out The Initial screening’. There is unnecessary capitalization of words.
4/ This is not an experimental study and labeling as ‘Result’ is not relevant.
5/ References in Tables are not in right order. Please modify.
6/Table 1 is too long and can be divided in to two or three tables. Classify the studies using a different parameter such as nature of the study, year of study, or study methodology used.
7/ The discussion section is more of a rewriting a literature review and a critical analysis on major study could be given as an opinion by authors.
Author Response
Please consider the following comments and suggestions in your revision.
Overall, the review paper is well written and it’s a comprehensive review which includes important studies in the field. However, it should the following points need to be incorporated.
1/ Abstract is very generic, and it doesn’t accurately reflect the entire contents in the review paper. Authors should include reflections why this review is important and major works should be highlighted.
We appreciate the reviewer’s feedback, acknowledging that the initial abstract was overly generic and did not adequately highlight the substantial findings. Therefore, a new version is now included, taking into account the journal format instructions (about 200 words maximum, a single paragraph, and following the style of structured abstracts, but without headings) (Page 1, Lines 24-39).
“In recent years, there has been growing interest in understanding the potential role of microbiota dysbiosis or alterations in the composition and function of human microbiota in the development of chronic rhinosinusitis with nasal polyposis (CRSwNP). This systematic review evaluated the literature on CRSwNP and host microbiota for the last ten years, including mainly nasal bacteria, viruses, and fungi, following the PRISMA guidelines, and using the major scientific publication databases. Seventy original papers, mainly from Asia and Europe, met the inclusion criteria, providing a comprehensive overview of the microbiota composition in CRSwNP patients and its implications for inflammatory processes in nasal polyps. This review also explores the potential impact of microbiota-modulating therapies for the CRSwNP treatment. Despite variability in study populations and methodologies, findings suggest that fluctuations in specific taxa abundance and reduced bacterial diversity can be accepted as a critical factor influencing the onset or severity of CRSwNP. These microbiota alterations appear to be implicated in triggering cell-mediated immune responses, cytokine cascade changes, and defects in the epithelial barrier. Although further human studies are required, microbiota-modulating strategies could become integral to future combined CRSwNP treatments, complementing current therapies that mainly target inflammatory mediators and potentially improving patient outcomes”.
2/ Please avoid abbreviations from the key word. And delete the phrase ‘Keywords (List three to ten). It is copied.
We appreciate the comment and apologize for the mistake. We have deleted abbreviations and “List three to ten” from keywords (Page 1, Lines 40-41).
3/ In the introduction section, what does it mean ‘The review includes both adult and children’s studies and excludes literature reviews? Why are literature reviews excluded from the study?
We appreciate the reviewer’s question. Literature reviews were excluded from our study because our main objective was to provide a precise, up-to-date, and unduplicated synthesis of the available information about CRSwNP and human microbiota from primary research publications. According to previous authors (DOI: 10.2106/JBJS.RVW.23.00077), reviews are excluded for several key reasons, such as avoidance of double-counting of studies, potentially introducing bias or lower-quality evidence, or inconsistencies and difficulties in synthesizing primary data. The available evidence has been identified, evaluated, and synthesized with a rigorous and standardized methodology with the aim that our work can be reproducible to a reasonable degree. Examples of previous systematic reviews that followed the same search strategy that did not include literature reviews are the following (DOI): 10.1080/03007995.2020.1815682, 10.3390/jpm13111583, and 10.3390/genes11040442.
However, multiple reviews have been included in both the introduction and discussion sections to provide contextual information to help in understanding the broader implications of results from primary studies, for example.
4/ Some phrases are not written appropriately such as ‘Three review authors carried out The Initial screening’. There is unnecessary capitalization of words.
We appreciate the reviewer’s comment and apologize for the typo. We have, therefore, modified the phrase indicated (Page 5, Line 190) and reviewed the text again.
5/ This is not an experimental study, and labeling as ‘Result’ is not relevant.
We appreciate the question. Notwithstanding, we have prepared our manuscript following the journal format instructions that indicate the following: “Structured reviews and meta-analyses should use the same structure as research articles and should ensure they conform to the PRISMA guidelines” (https://www.mdpi.com/journal/ijms/instructions#preparation).
Following the PRISMA reporting guidelines (DOI: 10.1136/bmj.n71), the results section (study selection, study characteristics, risk of bias in studies, results of individual studies, results of syntheses, reporting biases and certainty of evidence) must be included in a systematic review (https://www.prisma-statement.org/).
5/ References in Tables are not in right order. Please modify.
Again, following the journal format instructions, “references must be numbered in order of appearance in the text (including table captions and figure legends) and listed individually at the end of the manuscript”. Additionally, the data from the works in the primary tables were presented chronologically to facilitate the analysis of temporal trends and patterns.
6/Table 1 is too long and can be divided in to two or three tables. Classify the studies using a different parameter such as nature of the study, year of study, or study methodology used.
We totally agree that Table 1 is extensive. However, we consider that the relevant information included should not be displayed separately, following other systematic reviews with similar kinds of tables (DOIs: 10.3390/ijms23094494, 10.3390/genes11040442, 10.3390/ijms24054665). The articles included in the three primary tables of this manuscript have been included chronologically, with the earliest articles (2014) listed first and the most recent articles (2024) appearing last.
We believe that dividing Table 1 based on the study nature, year, or methodology used would substantially increase the number of tables and footnotes, reducing the manuscript’s clarity and adding unnecessary complexity. It is worth noting that, for example, dividing the table based on the methodology used would be particularly complex since multiple articles included different standard microbiological and molecular techniques.
However, the primary tables are now divided according to other systematic reviews published in the International Journal of Molecular Sciences, including the table headers (DOI: 10.3390/ijms23094494, 10.3390/ijms24054665, 10.3390/ijms24108632) (Table1: Pages 11-37, Lines 365-416; Table2: Pages 39-44, Lines 476-489; Table3: Pages 46-48, Lines 521-526). If the reviewer considers the separation of Table 1, we will prepare alternative tables, but we kindly suggest keeping it in the current format.
7/ The discussion section is more of a rewriting a literature review and a critical analysis on major study could be given as an opinion by authors.
We appreciate the reviewer’s insightful comment, but we have tried to provide a sophisticated and analytical discussion, meeting the expectations for a critical analysis of existing literature. As far as possible, we have identified and highlighted the most impactful studies of each main section, providing a detailed examination of their results and broader implications. Thus, the limitations of the evidence included and the review processes that potentially impact the final appreciation of the results included in the systematic review were initially discussed in line with the PRISMA 2020 statement, reflecting our commitment to a rigorous and sophisticated discussion.
However, we recognize the need to incorporate a more critical perspective on the key findings to enhance the discussion section. Therefore, we have revised the discussion section to include new paragraphs that provide our interpretation and perspective on the findings, discussing their potential implications for clinical practices and future research (Pages: 54-55, Lines 833-856). We believe this addition enhances the quality of our manuscript, aligning it more closely with the reviewer’s expectations. We extend our gratitude for the reviewer’s valuable feedback.
“6. Future Directions
A high abundance of several pathogen-related taxa and low microbial diversity, associated with changes in host's innate and adaptive immune response and defects in the epithelial barrier, seem characteristic of CRSwNP patients. However, the specific bacterial taxa changes and the associated immune response and cytokines profiles were not always consistent across studies. These observations allow speculation on future directions for microbiological diagnosis and treatment of CRSwNP. It seems conceivable that future research focused on the geographic and ethnic variations in host microbiota could provide insights into the influence of environmental and intrinsic factors on CRSwNP, thereby identifying commonly relevant microbial targets and region-specific considerations. In this context, further longitudinal studies that provide an overview of microbiota changes over time would help to clarify whether dysbiosis is a cause or effect of the disease and identify key microbial taxa preceding CRSwNP onset. Future studies should also investigate the roles of nasal viruses and fungi and gut microbiota, providing a more comprehensive view of the CRSwNP aetiology. Advanced omics technologies, such as metagenomics, transcriptomics, or metabolomics, could elucidate how gut microbiota influences nasal health and the mechanistic underlying microbiota, immune response, and barrier integrity.
Finally, the identification of specific microbiota profiles as potential diagnostic biomarkers could lead to non-invasive methods for early detection and monitoring of CRSwNP. Well-designed clinical trials with extensive CRSwNP patient recruitment could enable researchers to develop appropriate and personalized microbiota-modulating strategies adapted to specific microbiota profiles, enhancing treatment efficacy and minimizing adverse effects”.

Round 2
Reviewer 1 Report
Comments and Suggestions for Authors
The authors have significantly improved the scientific impact and quality of this manuscript.